# Human glia can both induce and rescue aspects of disease phenotype in Huntington disease

Abdellatif Benraiss[1], Su Wang[1], Stephanie Herrlinger[1], Xiaojie Li[1], Devin Chandler-Militello[1], Joseph Mauceri[1], Hayley B. Burm[1], Michael Toner[1], Mikhail Osipovitch[2], Qiwu Jim Xu[1], Fengfei Ding[1], Fushun Wang[1], Ning Kang[1], Jian Kang[3], Paul C. Curtin[4], Daniela Brunner[4], Martha S. Windrem[1], Ignacio Munoz-Sanjuan[5], Maiken Nedergaard[1,2] & Steven A. Goldman[1,2,6]

The causal contribution of glial pathology to Huntington disease (HD) has not been heavily explored. To define the contribution of glia to HD, we established human HD glial chimeras by neonatally engrafting immunodeficient mice with mutant huntingtin (mHTT)-expressing human glial progenitor cells (hGPCs), derived from either human embryonic stem cells or mHTT-transduced fetal hGPCs. Here we show that mHTT glia can impart disease phenotype to normal mice, since mice engrafted intrastriatally with mHTT hGPCs exhibit worse motor performance than controls, and striatal neurons in mHTT glial chimeras are hyperexcitable. Conversely, normal glia can ameliorate disease phenotype in transgenic HD mice, as striatal transplantation of normal glia rescues aspects of electrophysiological and behavioural phenotype, restores interstitial potassium homeostasis, slows disease progression and extends survival in R6/2 HD mice. These observations suggest a causal role for glia in HD, and further suggest a cell-based strategy for disease amelioration in this disorder.

[1] Center for Translational Neuromedicine, University of Rochester Medical Center, Rochester, New York 14642, USA. [2] Center for Basic and Translational Neuroscience, University of Copenhagen, 2200 Copenhagen, Denmark. [3] Department of Anatomy and Cell Biology, New York Medical College, Valhalla, New York 10595, USA. [4] Psychogenics, Inc., Tarrytown, New York 10591, USA. [5] CHDI Foundation and CHDI Management, Inc., 6080 Center Drive, Suite 100, Los Angeles, California 90045, USA. [6] Neuroscience Center, Rigshospitalet, 2100 Copenhagen, Denmark. Correspondence and requests for materials should be addressed to S.A.G. (email: steven_goldman@urmc.rochester.edu or goldman@sund.ku.dk).

Glial pathology may contribute to a broad set of neurodegenerative and neuropsychiatric diseases traditionally considered disorders of solely neuronal dysfunction[1–5]. Huntington's disease (HD) is a prototypic neurodegenerative disorder, characterized by abnormally long CAG repeat expansions in the first exon of the Huntingtin gene. The encoded polyglutamine expansions of mutant huntingtin (mHTT) protein disrupt its normal functions and protein–protein interactions, ultimately yielding widespread neuropathology, most rapidly evident in the neostriatum. Yet, despite the pronounced loss of neostriatal medium spiny neurons (MSNs) in HD, and evidence of glial dysfunction[6,7], few studies have investigated the specific contribution of glial pathology either to striatal neuronal dysfunction in HD, or more broadly, to disease phenotype. Our lack of understanding of the role of glial pathology in HD has reflected the lack of *in vivo* models that permit the separate interrogation of glial and neuronal functions in HD, particularly so in humans. Indeed, this gap in our knowledge is especially concerning in light of the marked differences between human and rodent glia; human astrocytes are larger and more structurally complex than rodent glia, and influence the actions of vastly more synapses within their geographic domains[8,9]. Accordingly, mice neonatally engrafted with human glial progenitor cells (hGPCs), which develop brains chimeric for human astroglia and their progenitors[10], exhibit substantially enhanced activity-dependent plasticity and learning[11]. Yet the relatively greater role of human astrocytes in neural processing suggests the potential for glial pathology to wreck especial havoc within human neural circuits, with attendant implications for the human neurodegenerative disorders.

In this study, we identified a specific role for human striatal glia in the pathogenesis of HD, by comparing the behaviour and MSN physiology of human glial chimeric mice xenografted at birth with mutant HD-expressing human hGPCs to their normal HTT hGPC-engrafted controls. In particular, we first compared the motor behaviour of immunodeficient mice neonatally xenografted with hGPCs produced from mutant HD (48 CAG) human embryonic stem cells (hESCs), to that of controls engrafted with hGPCs derived from a sibling line of unaffected hESCs (18 CAG). We found that the HD hESC GPC-engrafted mice manifested impaired motor learning relative to control hGPC-engrafted mice. On that basis, we then used lentiviral transduction of astrocyte-biased hGPCs derived from second trimester human forebrain, to generate lines of hGPCs carrying either normal (23 CAG) or HD (73 CAG) repeats. To that end, we sorted the fetal tissue samples for CD44, a hyaluronic acid receptor ectodomain expressed by astrocyte-biased glial progenitor cells[12], and infected the CD44-immunoselected cells with the lentiviral mHTT vectors. We then assessed the effects of mouse striatal implantation of these human mHTT glia on local neuronal physiology, and found that the striatal neurons of mHTT (73 CAG) glial-engrafted mice exhibited increased neuronal input resistance and excitability, relative to those of mice engrafted with normal HTT (23 CAG)-transduced striatal glia.

On that basis, we then asked if neonatal chimerization with normal glia might delay disease progression in R6/2 transgenic HD mice[13]. We found that the substantial replacement of diseased striatal glia with wild-type (WT) CD44[+] human glia indeed resulted in a slowing of disease progression, and a corresponding increment in survival in transplanted R6/2 mice. This was associated with a transplant-associated fall in neuronal input resistance, and a corresponding drop in interstitial K[+] in the R6/2 striatum. Together, these studies suggest both a critical role for glial pathology in the progression of HD, and the potential for glial cell replacement as a strategy for its treatment.

## Results

**Glia were generated from hESCs expressing mHtt**. We previously developed a high-efficiency protocol for generating GPCs and their derived astroglia and oligodendrocytes from both hESCs and induced pluripotential cells[14]. Neonatal engraftment of these cells into immunodeficient mice yields human glial chimeric mice, in which substantially all GPCs and a large proportion of astrocytes are of patient-specific, human donor origin. Using this approach, we first sought to generate GPCs from huntingtin mutant pluripotential cells, and to then establish human glial chimeras with those cells, as a means of assessing the specific effects of human huntingtin mutant glia on striatal function.

To that end, we used huntingtin mutant hESCs, the GENEA 20 line bearing a 48 CAG repeat expansion in the first exon of the HTT gene along with a normal 17 CAG allele, as well as its matched sibling control, GENEA 19, which has normal 18 and 15 CAG repeat lengths in exon 1 (ref. 15). These lines were derived from blastocysts produced from the same parents, and were thus fraternal twins. We then induced GPCs from these hESC lines, using our described protocol[14]. When collected after an average *in vitro* propagation of 200 days of glial induction (range of 160–240), an average of $56.0 \pm 4.6\%$ of normal (GENEA 19) and $45.8 \pm 7.0\%$ of huntingtin mutant (GENEA 20) cells expressed the bipotential astrocyte–oligodendrocyte progenitor marker PDGFαR/CD140a (ref. 16). The remainder were almost entirely CD44[+]/CD140a[−] cells, which typify astroglia and their progenitors[12]. Immunostaining revealed that <1% of cells expressed either the neuronal antigens HuC/D or MAP2, and pluripotency-associated gene expression was undetectable by either immunocytochemistry or quantitative PCR. Thus, the grafted cells were comprised almost entirely of CD44-defined astroglial progenitors and bipotential oligodendrocyte–astrocyte GPCs. The GENEA 20- and GENEA 19-derived glia were neonatally engrafted bilaterally into the neostriata of rag1[−/−] immunodeficient mice ($n = 38$ and 35, respectively), to establish mHtt human glial chimeras and their normal human glial controls.

**Chimerization yielded host colonization by mHtt[+] human glia**. On weaning, the human glial chimeric mice were then randomly assigned to matched groups for either serial analysis of their motor performance by rotarod, or for serial sacrifice for histological analysis as a function of age. Histological analysis revealed that the striata of these mice rapidly and efficiently engrafted with donor hESC-derived hGPCs (Fig. 1a,b and Table 1). The donor cells first expanded to pervade the host striata as persistent hGPCs, in part replacing the resident murine GPCs in the process (Fig. 1c–f). A fraction of the donor cells then differentiated as astroglia, especially so in striatal white matter tracts. Fibrous astrocytes appeared early, and were arrayed densely within striatal white matter tracts by 6–8 weeks after neonatal graft, whereas striatal protoplasmic astrocytes appeared later, and were first apparent in significant numbers only by 12 weeks (Fig. 1g,h). Over the weeks thereafter, the host striatal hGPCs were substantially replaced by human donor cells, whether by HD hESC-derived hGPCs or their normal sibling-derived hGPCs; in each case, hGPCs were typically the dominant population by 20 weeks, and few if any murine GPCs remained in any of the engrafted striata after 40 weeks (Fig. 1c,d). Transplanted cells did not differentiate into neurons, as evidenced by their lack of expression of either MAP2 or NeuN, two distinct markers of mature neuronal phenotype. No evidence of tumour formation or aberrant differentiation of these hESC-derived GPCs was noted in any of the mice in this study.

**Motor performance was impaired in mHtt glial chimeras**. Among the 109 mice assigned to rotarod assessment of motor

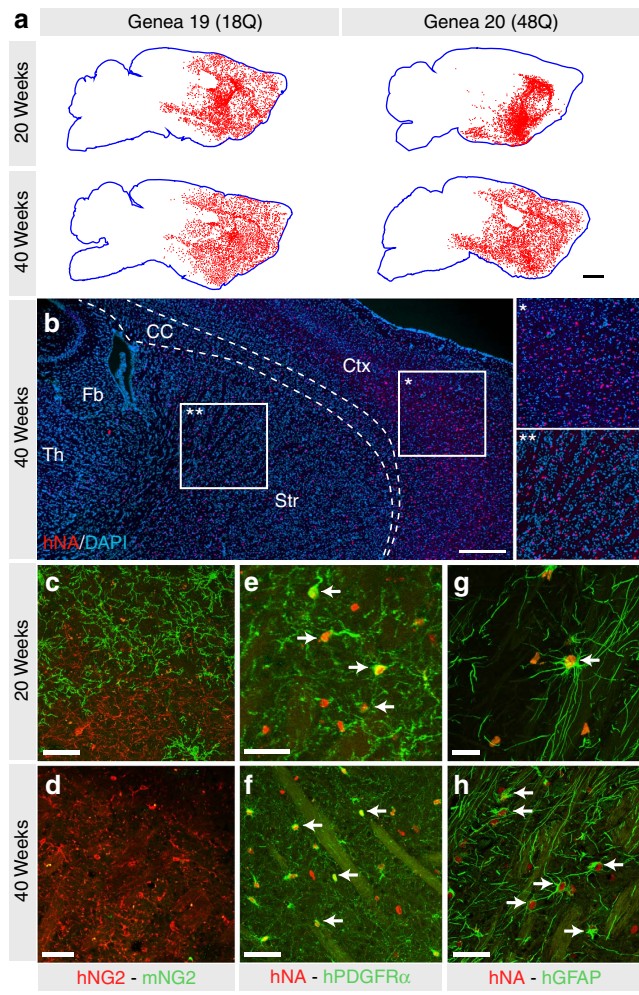

**Figure 1 | Mice may be generated with striata chimeric for human HD ESC-derived glia.** The striata of neonatally engrafted rag1 null mice were efficiently colonized with donor hESC hGPCs, a proportion of which differentiated as astroglia in both striatal grey and white matter. (**a**) Mice engrafted with hESC GPCs expressing either normal Htt (GENEA19; 18Q) or mHtt (GENEA20, 48Q; a sibling to GENEA19) manifested striatal chimerization by 20 weeks of age, which was denser at 40 weeks. Mice engrafted with GENEA20-derived glia (48Q) manifested striatal chimerization analogous to that of GENEA19-derived normal HTT (18Q) glia. (**b**) GENEA19-derived glia identified by their expression of the human-specific nuclear antigen (hNA; red) interspersed with host cells (DAPI, blue), revealing extent of striatal and cortical human glial chimerization at 40 weeks. (**c,d**) GENEA19 hGPC-engrafted striatal sections at 20 (**c**) and 40 (**d**) weeks post graft, stained for human and mouse NG2, showing the progressive domination of the striata by human NG2-defined GPCs. (**e,f**) 20 (**e**) and 40 (**f**) week post-graft striata, stained for hNA (red) and human PDGFRA (green) similarly showing the progressive domination of the striata by hGPCs. (**g,h**) GENEA19 hGPC-engrafted striata at 20 (**g**) or 40 (**h**) weeks, stained for hNA (red) human GFAP (green), showing the maturation and age-dependent increase in fibre complexity of human astroglia in the host striatum. Arrows indicate graft-derived OPCs (**e,f**) and astrocytes (**g,h**). Scale bars, 1 mm (**a,b**); 50 μm (**c–g**); 25 μm (**h**). DAPI, 4,6-diamidino-2-phenylindole.

performance, those chimeric for HD mHTT hESC (GENEA20)-derived glia manifested significantly slowed motor learning compared with littermates chimerized with normal HTT GPCs (GENEA19). In particular, the GENEA20-derived mHtt glial chimeras manifested significant decrements in motor coordination relative to three independent control groups that included: (1) GENEA19 GPC-derived chimeric controls; (2) uninjected controls; and (3) saline-injected controls (Fig. 2). The difference between mHTT and all control glial chimeras was evident by 12 weeks of age, and persisted through 36 weeks of observation, with no significant improvement in the performance of the mHTT GPC-engrafted mice during that 24-week period (Supplementary Table 2). The relative lack of improvement in rotarod performance by the mHTT hGPC-engrafted mice suggested an mHTT glial-mediated deficit in motor learning, as well as in motor performance, that became increasingly manifest with age and maturation.

**MSNs were hyperexcitable in the presence of mHtt glia.** To better understand the physiological basis for the relatively impaired motor performance of mHtt glial-engrafted mice, we next asked whether chimerization with mHtt-expressing glia influenced the physiology of MSNs. To that end, we established striatal glial chimeras in otherwise WT immunodeficient mice, via neonatal intrastriatal injection of mHtt-expressing human fetal glia. For this purpose, we used mHTT-transduced fetal tissue-derived hGPCs rather than HD hESC-derived GPCs, so as to assess the effects of mHtt bearing longer CAG repeats than the 48Q mHtt expressed by GENEA 20-derived hGPCs. We postulated that longer CAG repeat expansions would accelerate glial pathology, and thus potentiate detection of paracrine neuronal dysfunction at the relatively young ages and compressed experimental time frames used in this study. To that end, we isolated hGPCs from 18- to 20-week human fetal forebrain, using immunomagnetic sorting directed against CD44, which as noted is highly expressed by astrocyte-biased glial progenitor cells[12]. We then transduced these cells with a lentiviral vector encoding the first exon of the HTT gene bearing either mutant (73Q) or normal (23Q) huntingtin, each upstream to an enhanced green fluorescent protein (EGFP) reporter, and then injected the transduced cells into the striata of neonatal rag1$^{-/-}$ immune-deficient mice. The mice were killed 12 weeks later and striatal slices were prepared; human GFP$^+$ glial-rich regions were imaged by two-photon microscopy, and their resident striatal neurons patch clamped using previously described methods[17]. Subsequent histology and immunolabelling confirmed the dense engraftment of the recorded striata with human donor cells, in both the Q23 and Q73 mHtt hGPC-engrafted striata, whose extents of donor cell engraftment were indistinguishable at the 12-week time point at which recordings were obtained (Supplementary Fig. 1). Of note, whereas the distributions of Q23 and Q73 mHtt-transduced glia did not differ in engrafted chimeras (Supplementary Fig. 1D), and their relative densities similarly did not significantly differ (Supplementary Fig. 1E), the Q73 mHtt glia could be recognized by cytoplasmic Htt aggregates, *in vivo* as well as in culture (Supplementary Fig. 1A–C,H); the Q23-transduced controls manifested no such aggregate formation.

Physiologically, neurons in striata engrafted with 73Q mHtt glia manifested significantly higher input resistance relative to those engrafted with either 23Q HTT- or EGFP-only transduced control glia, and required significantly fewer current injections to fire action potentials relative to control glia-engrafted mice (Fig. 3a,b; also Supplementary Fig. 2). The higher input resistance of these neurons was manifest in their current–voltage (I–V) curves as well (Fig. 3c), and suggested the significant relative hyperexcitability of striatal neurons in an mHtt-glial environment (Fig. 3d). This was also reflected by the less-negative resting membrane potentials of striatal neurons recorded in 73Q hGPC-engrafted striata ($-75.3 \pm 0.53$ mV, mean ± s.e.m.),

**Table 1 | Engraftment of hESC-derived GPCs in normal Htt immunodeficients.**

| Cell type | Sacrifice | % GFAP$^+$ | % Olig2$^+$ | Total donor cells | humanNA$^+$/mm$^3$ striatum |
|---|---|---|---|---|---|
| GENEA19 (18Q) | 20 weeks ($n=3$) | $2.1 \pm 0.6$ | $71.8 \pm 19.4$ | $74,173 \pm 14,305$ | $21,000 \pm 3,608$ |
| | 40 weeks ($n=4$) | $1.7 \pm 0.5$ | $82.5 \pm 8.6$ | $42,807 \pm 6,991$ | $9,335 \pm 1,341$ |
| GENEA20 (48Q) | 20 weeks ($n=3$) | $2.3 \pm 0.4$ | $56.4 \pm 7.5$ | $42,520 \pm 8,792$ | $10,843 \pm 3,323$ |
| | 40 weeks ($n=4$) | $2.2 \pm 0.7$ | $72.7 \pm 6.3$ | $80,798 \pm 7,131$ | $16,126 \pm 380$ |

hNA, human nuclear antigen.
Data presented as means ± s.e.m.'s.

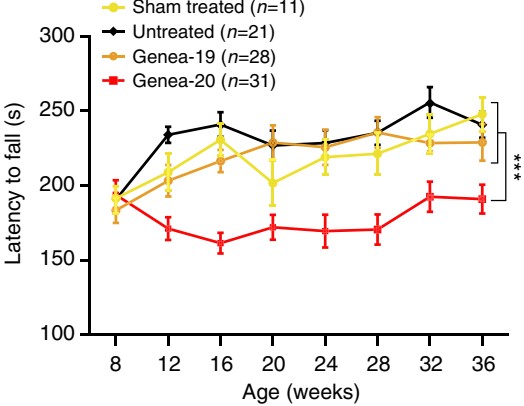

**Figure 2 | HD ESC-derived glial chimeras exhibit impaired motor coordination.** Mice engrafted with GENEA20-derived glia expressing mHtt demonstrated significantly impaired motor coordination compared with littermates chimerized with either GENEA19-derived normal HTT GPCs or control mice (both sham-treated and untreated). In particular, the mHTT glial chimeras (GENEA20, with 48Q; $n=31$) manifested significant age-dependent decrements in motor coordination relative to their normal HTT (GENEA19, 18Q; $n=28$) hESC GPC-derived chimeric controls, as well as relative to sham-treated ($n=11$) and untreated controls ($n=21$). Two-way ANOVA revealed both a significant treatment effect ($F(3, 593) = 39.6$, ***$P < 0.0001$) and time effect ($F(7, 593) = 5.47$, $P < 0.0001$); mean ± s.e.m.

relative to the 23Q hGPC-engrafted ($-79.4 \pm 0.93$ mV), untransduced hGPC-engrafted ($-80.5 \pm 0.51$ mV) and unengrafted ($-81.1 \pm 0.94$ mV) controls ($F[3,35] = 17.0$; $P < 0.0001$ (Fig. 3e); sample sizes and *post hoc* test results noted in 3E, statistical summary in Supplementary Table 3A). The I–V curves of the recorded MSNs showed typical inwardly rectifying currents, which were also seen in the input resistance at hyperpolarization and depolarization currents (Fig. 3f; statistical summary in Supplementary Table 3B). Interestingly, despite their higher $V_m$ and hyperexcitability in response to current injection, neither the frequency (Fig. 3f,g) nor amplitude (Fig. 3f,h) of either the spontaneous excitatory postsynaptic currents (sEPSCs) or miniature EPSCs of striatal neurons within 73Q glial chimeric striata differed significantly from those of striatal neurons in 23Q-engrafted or unengrafted controls (Fig. 3g,h).

**Colonization by normal glia slowed disease course in HD mice.** Since engraftment of normal striata with mHtt-expressing GPCs impaired striatal neuronal function and physiology, we next asked whether the reverse might be true, that is, if neonatal engraftment of the HD striata with normal glia might rescue aspects of HD phenotype. To this end, we engrafted normal human GPCs into the striata of newborn R6/2 (120 CAG) mice[13], which transgenically express a mutant exon 1 of the HTT gene, and

typically die by 20 weeks of age. For this experiment, astrocyte-biased GPCs were isolated from 18- to 22–week-gestational age fetal human brain using magnetic activated cell sorting targeting CD44, as noted above (Supplementary Fig. 3A)[12]. The CD44-sorted cells were then transplanted into the striata of newborn R6/2 × rag1$^{-/-}$ mice, using an injection protocol previously described for use in neonatal callosal injection[18] (Supplementary Fig. 3B–E), but instead targeting the striata. Striatal engraftment of the R6/2 mice by CD44-sorted hGPCs was robust (Fig. 4a,b), and achieved densities of >15,000 human cells per mm$^3$ by 16 weeks of age (Fig. 4c and Table 2), with substantial replacement of resident mouse HD astroglia with normal HTT-expressing human counterparts, as we have previously reported in WT murine hosts[10]. The human CD44-sorted glia integrated as both astrocytes (Fig. 4d and Table 2) and as persistent GPCs (Fig. 4e,f and Table 2), but not as neurons (Fig. 4g). Importantly, the integrated human cells did not manifest detectable HTT aggregates; the staining patterns of HTT and human nuclear antigen were always entirely non-overlapping (Fig. 4h). As such, we saw no evidence of HTT protein transmission from host to donor cells over the time frame studied. While there was a net weight loss in diseased mice as function of time (8- and 16-week-old mice), no change of weight as function of engraftment was noticed (age effect: $F[1,36] = 8.40$; $P < 0.01$; treatment effect: $F[1,36] = 0.12$; $P > 0.05$; two-way analysis of variance (ANOVA), statistical summary in Supplementary Table 4).

The hGPC-engrafted chimeric R6/2 mice displayed significantly delayed motor deterioration relative to their untreated controls, as assessed by their performance on a constantly accelerating rotarod. Linear regression revealed that the rate of motor deterioration was significantly slowed in the hGPC engrafted mice, relative to untreated and sham-treated controls ($F = 4.8$ [2, 124 d.f.]; $P < 0.001$) (Fig. 5a and Supplementary Table 5).

On that basis, we next asked if the performance enhancement associated with engraftment by normal glia might be sufficient to influence the survival of R6/2 (120 CAG) mice. We found that R6/2 (120Q) × rag1$^{-/-}$ mice whose striata were neonatally transplanted with normal human glia survived significantly longer than unengrafted mice, with a mean increase in lifespan of 12 days (hGPC-engrafted, $n=29$; untreated, $n=28$; $P < 0.01$, Mantel–Cox Log-rank test; Fig. 5b). This survival effect was no different between males and females among the in hGPC-engrafted R6/2 mice (statistical summary in Supplementary Table 6).

The functional benefits of R6/2 striatal chimerization with normal glia were accompanied by preservation of striatal structure as well. Stereological assessment of gross striatal volume revealed that CD44$^+$ hGPC-engrafted mice manifested significantly less striatal involution than their unengrafted controls (Fig. 5c). This decreased rate of striatal atrophy suggested a relative preservation of striatal neuropil, which was both substantial and statistically significant by 20 weeks of age ($F(2, 25) = 12.84$, $P = 0.0001$ by

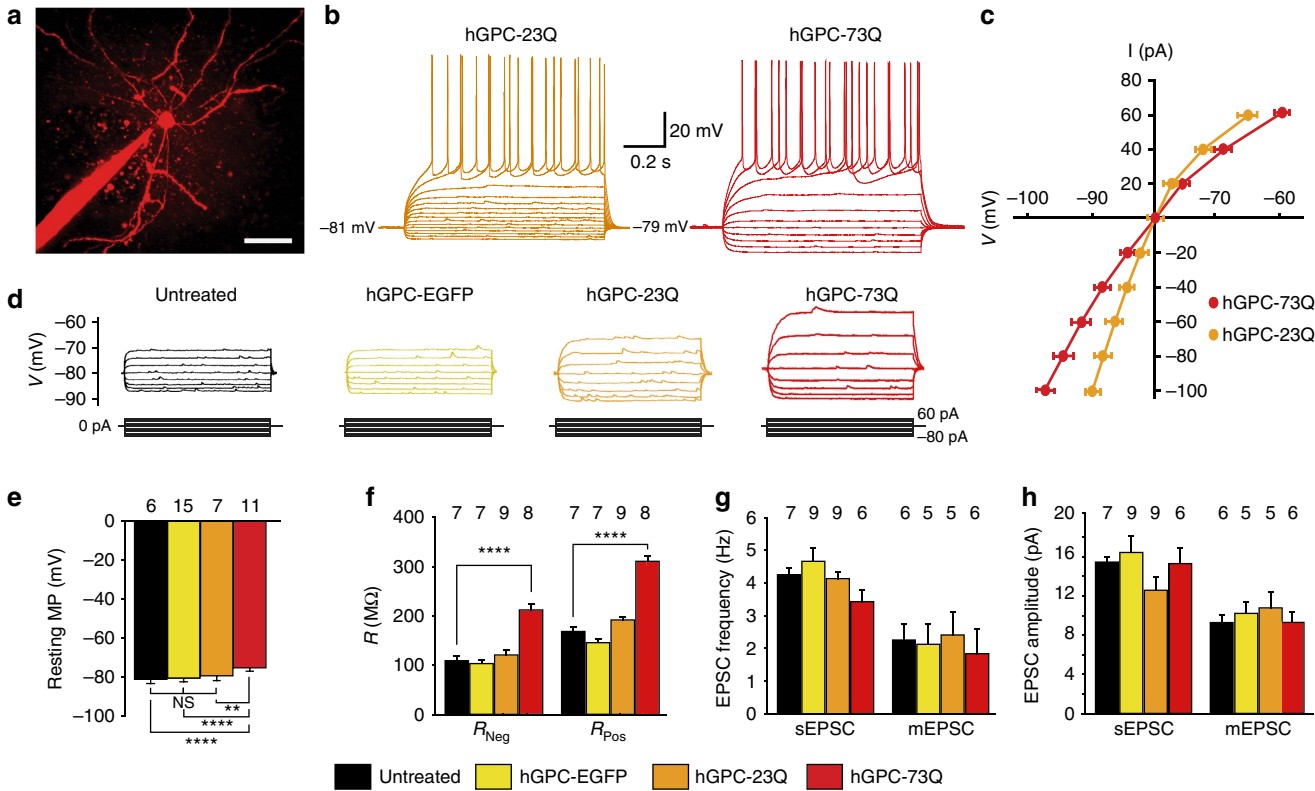

**Figure 3 | Striatal neurons are hyperexcitable in mice chimerized with mHTT-transduced hGPCs.** Human glial chimeric striata were established with fetal human glia transduced to overexpress mHTT. (**a**) Example of a host striatal neuron, filled with Alexa-594 after recording, surrounded by EGFP-tagged donor-derived glia. (**b**) Representative action potentials recorded in response to current injection in host neurons in striata chimerized with 73Q mHTT ($n = 8$)- and 23Q HTT ($n = 12$)- expressing hGPCs. Striatal neurons of mice engrafted with 73Q mHTT human glia required significantly fewer current injections to achieve voltage thresholds for firing, than did those engrafted with 23Q HTT-transduced or either EGFP-only ($n = 8$) transduced or untreated ($n = 8$) control glia. (**c**) Current–voltage curves (I–V curves) of neurons in 23Q HTT- and 73Q mHTT-expressing hGPC-engrafted mice, reflect the typical inwardly rectifying currents of MSNs. (**d**) Representative traces of injected current (20-pA steps)-induced voltage changes are shown for the four treatment groups. The waveforms of stimulus injection are shown below the tracings. (**e**) The relative hyperexcitability of striatal neurons in the mHtt glial environment was also reflected by the higher resting membrane potential of those neurons, relative to both Q23 hGPC-engrafted and unengrafted controls. (**f**) The input resistances at negative current injection ($R_{Neg}$, $-40$ pA hyperpolarization currents) were compared with those with positive current injection ($R_{Pos}$, 40 pA depolarization currents), and confirmed the higher input resistance of striatal neurons in 73Q glial chimeras, relative to both 23Q and GFP control glia-engrafted mice. (**g,h**) Comparison of frequency (**g**) and amplitude (**h**) of sEPSCs and miniature EPSCs (mEPSCs). Despite their relative hyperexcitability, striatal neurons within 73Q glial chimeric striata manifested sEPSC frequencies and amplitudes that did not differ significantly from those of either 23Q glial-engrafted or unengrafted striatal neurons. Scale bar, 50 µm (**a**); (**e,f**) \*\*$P < 0.01$; \*\*\*\* $P < 0.0001$ by ANOVA with *post hoc* t tests; means ± s.e.m.

two-way ANOVA). By that point, the mean striatal volume of hGPC-engrafted R6/2 mice was significantly larger than that of unengrafted R6/2 mice ($P = 0,007$ with Tukey's multiple comparison tests; Supplementary Table 7), and only marginally less than that of WT mice (WT mice: $7.5 \pm 0.1$ mm$^3$; R6/2 untreated: $4.7 \pm 0.4$ mm$^3$; R6/2-hGPC: $6.3 \pm 0.6$ mm$^3$ (mean ± s.e.m.)). Thus, neonatal hGPC transplantation was associated with significantly increased striatal volumes in R6/2 mice, which was substantial and significant by 20 weeks of age (Fig. 5c).

**Normal glia improved the behaviour and cognition of R6/2 mice.** We next asked whether striatal engraftment with CD44-defined hGPCs was sufficient to improve cognitive and motor function by HD mice. To this end, we neonatally transplanted a large cohort of R6/2 mice with bilateral intrastriatal injections of $50 \times 10^4$ CD44-sorted hGPCs (Supplementary Table 1). Both the transplanted and control mice were then sent to an independent contract research organization, Psychogenics, Inc., which used a multimodal behavioural platform to compare the behavioural

repertoires of the hGPC-engrafted and control mice, the identities to which Psychogenics staff were blinded. This platform included two proprietary batteries, called Smartcube, a battery of cognitive end points[19], and Neurocube, a battery of motor functional end points[19]. A third separate test, the procedural water T-maze, was also included. The individual functional elements of these testing paradigms were represented in the aggregate by a single interpolated value, which permitted group comparisons across all endpoints with a form of principal component analysis (PCA).

PCA revealed that across test modalities, the R6/2 mice could be readily distinguished from both their WT uninjected and sham-injected controls (Fig. 6a–d). PCA further revealed that hGPC transplants were associated with at least a partial restoration of normal behavioural end points in multiple domains (Supplementary Figs 4 and 5). While the component composition of the overall PCA is proprietary, single-behavioural end points could be identified and extracted *post hoc*; these are reported separately in Supplementary Figs 4 and 5, which show behavioural tests in which hGPC-engrafted R6/2 mice are notably less impaired than their unengrafted controls.

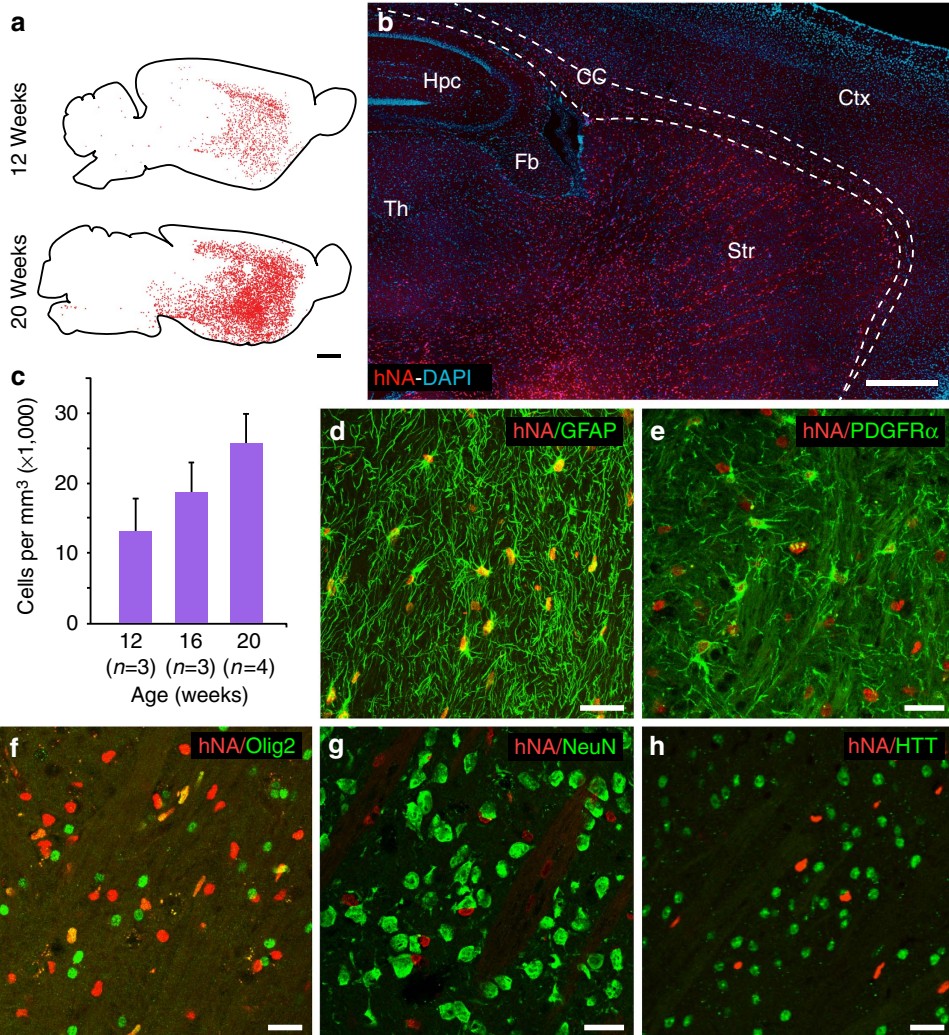

**Figure 4 | CD44-sorted hGPCs colonized and replaced endogenous glia within the R6/2 × rag1$^{-/-}$ striatum.** Striatal engraftment of the R6/2 mice by CD44-sorted hGPCs was robust and dense. (**a,b**) Fetal derived cells expanded to colonize the striata and ventral forebrain of engrafted mice by 20 weeks. (**c**) Donor-derived cells in the striata of transplanted mice increased as a function of time (means ± s.e.m.). (**d–g**) By 20 weeks after neonatal graft, the donor hGPCs (human nuclear antigen, red) integrated as astrocytes (**d**; GFAP, green) or persisted as GPCs (**e,f**; PDGFαR and olig2, green), but did not give rise to neurons; no overlap was ever seen of hNA and NeuN expression (**g**; NeuN, green). (**h**) Resident human glia did not manifest detectable nuclear Htt aggregates, as assessed by EM48 immunostaining; the staining patterns of host Htt and donor human nuclear antigen were always entirely non-overlapping (**h**). Scale bars, 1 mm (**a,b**); 25 μm (**d–h**).

**Table 2 | Engraftment of CD44$^+$ GPCs in R6/2 x rag1$^{-/-}$ mice.**

| Survival time | % GFAP$^+$ | % Olig2$^+$ | Total cells | hNA$^+$ per mm$^3$ striatum |
|---|---|---|---|---|
| 20 weeks (n=4) | 1.7 ± 0.3 | 45.2 ± 7.8 | 77,756 ± 21,000 | 16,651 ± 3,694 |

hNA, human nuclear antigen.
Data presented as means ± s.e.m.'s.

groups (36%, $P<0.02$; and 19%, $P=0.068$, respectively). Analysis of the top features that contributed to the disease signatures showed that at 8 weeks of age, sham-treated R6/2 mice were somewhat hyperactive compared with sham-treated WT mice, showing increased sniffing/scanning ($P=0.0003$), locomotion ($P=0.0083$) and rearing ($P=0.0022$). Some of these changes were attenuated in hGPC-treated R6/2 mice, which exhibited less locomotion ($P<0.0001$) and rearing ($P=0.0003$) relative to sham-treated R6/2 mice. At 11 weeks of age disease-associated hyperactivity subsided, with no significant differences between sham-treated WT and R6/2 in locomotion, rearing or scanning/sniffing. At both 8 and 11 weeks, R6/2 mice groomed less than WT mice ($P=0.0003$), a phenotype not rescued by hGPC. Overall, these results are consistent with a prodromal hyperactive phase described in studies of young R6/2 and R6/1 mice[20–22], which we found here to be tempered by hGPC treatment.

This analysis first revealed that disease-associated hyperactivity in R6/2 mice was moderated by neonatal hGPC graft. SmartCube showed a significant difference between sham-treated WT and R6/2 mice at 8 (91%, $P<0.0002$) and 11 weeks (92%, $P<0.00001$) of age (Supplementary Fig. 4). At both ages, relative improvement was noted as a result of hGPC treatment, although this effect failed to achieve significance in the older of the two R6/2

Our analysis next revealed that age-dependent gait deficits in R6/2 were partially rescued by hGPC treatment. NeuroCube analysis revealed that by 11 weeks of age, R6/2 mice manifested a

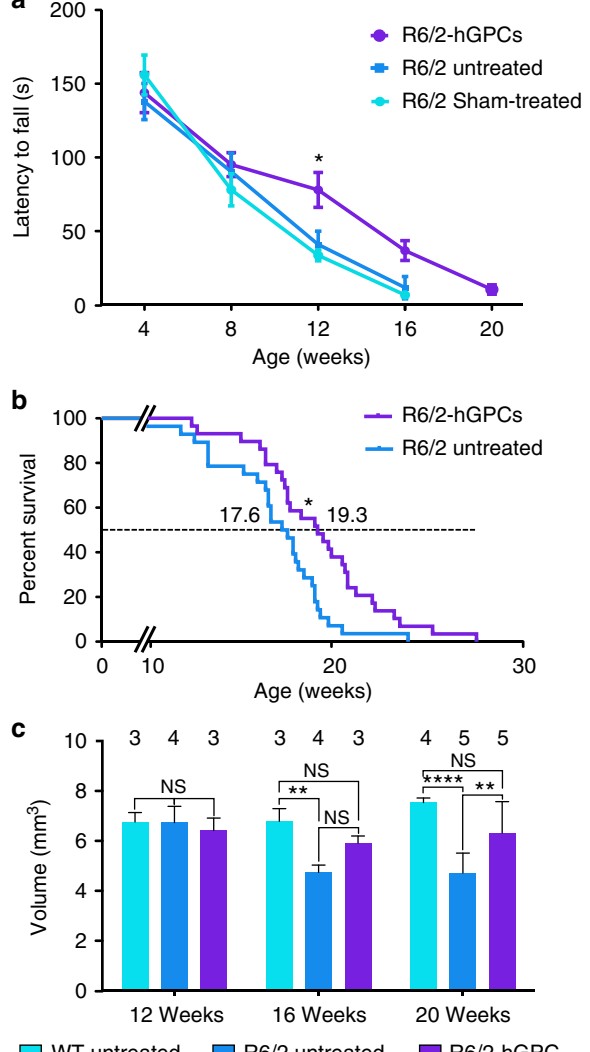

**Figure 5 | Chimerization with normal glia slows motor loss and extends survival of R6/2 mice.** (**a**) Linear regression revealed that the rate of rotarod-assessed motor deterioration of R6/2 mice was significantly slower in mice engrafted with hGPCs ($n=15$) than in untreated mice (sham-treated, $n=11$; untreated, $n=10$) ($F(3,608)=41.87$; $P<0.001$). (**b**) R6/2 (120Q) × rag1$^{-/-}$ mice whose striata were engrafted with human GPCs survived significantly longer than unengrafted mice ($n=29$ hGPC-engrafted; $n=28$ untreated; $P<0.01$ by Mantel–Cox Log-rank test). (**c**) Striatal volumes were estimated stereologically (Stereo Investigator, MicroBrightfield). hGPC-engrafted R6/2 mice manifested larger striatal volumes than unengrafted R6/2 mice by 16 weeks of age, which were restored to levels no different than those of WT controls. Means ± s.e.m.; $**P<0.01$ and $****P<0.0001$ by one-way ANOVA with *post hoc t*-tests.

respectively). Interestingly, hGPC treatment reduced body movement variability in both WT and R6/2 groups, an HD-independent effect ($P=0.003$).

In addition to the SmartCube and NeuroCube analyses of these mice, we also assessed their performance in the water T-maze, and found that the performance of R6/2 mice in this test was improved markedly by hGPC transplant. In this procedural water T-maze test, mice need to learn to swim to a side platform. We found that untreated 9-week-old R6/2 mice chose the correct side less frequently ($P<0.0001$) and reached the platform more slowly ($P<0.0001$) than WT mice (either sham-injected or hGPC-injected) Fig. 6e–g. In contrast, neonatal intrastriatal hGPC engraftment partially rescued this deficit, in that chimeric R6/2 mice performed better than their sham-operated R6/2 counterparts, making more correct choices in the initial session ($P=0.007$), and with more mice reaching criterion during initial training ($P=0.016$; Fig. 6e,f). At 13 weeks of age, deficits in choice accuracy were again apparent in sham-treated R6/2 mice relative to WT mice ($P<0.0001$), and in the percentage of mice reaching criterion ($P<0.0001$). Choice accuracy was improved in hGPC-treated R6/2 (relative to sham-treated R6/2) by the third and forth sessions ($P<0.02$). The latency to reach the escape platform was also significantly improved by hGPC treatment in the last three sessions, suggesting not only a cognitive but also a motor improvement (Fig. 6g).

**R6/2 MSN physiology was altered by normal glia.** To assess the physiological basis of the improved function and survival of R6/2 mice engrafted with human glia, we next asked whether chimerization with normal glia influenced the physiology of resident R6/2 striatal neurons. This experiment, examining the effects of a normalized glial environment on HD MSNs, provided a corollary to our prior experiment, in which normal MSNs were evaluated in an HD glial environment. To this end, 19 immunodeficient rag1$^{-/-}$ × R6/2 (120 CAG) newborns were either engrafted ($n=8$) or not ($n=11$) with CD44-sorted hGPCs; 12 weeks later, they were killed and slice preparations taken, and MSNs patch-clamped. In addition, 18 WT × rag1$^{-/-}$ mice were assessed, 7 of which had been neonatally engrafted with CD44-sorted hGPCs and 11 of which were unengrafted controls. Successful engraftment of the recorded striata by human donor cells was verified histologically after recording. Mice that displayed poor engraftment, defined as <1,000 human nuclear antigen$^{+}$ cells per mm$^{3}$, were removed from the study (versus an average of >10,000 cells per mm$^{3}$ in successful grafts; see Tables 1 and 2).

Whole-cell voltage-clamp recording from both WT and R6/2 striatal neurons showed inward rectifier currents as the membrane potential was between −115 and −45 mV, and more rectification was found in R6/2 neurons (Fig. 7a). Similarly, with the current-clamp configuration, we found that the input resistance $R_{input}$ of R6/2 striatal neurons was significantly higher than that of their WT controls, as has been previously reported[23,24]. Significantly though, we found that the R6/2 neuron $R_{input}$ in the range of positive membrane potentials was lower in the presence of engrafted normal human CD44-derived glia (Fig. 7b,c and Supplementary Table 8A). These results indicate that the intrinsic excitability of striatal neurons in R6/2 mice was increased relative to that of WT neurons, and could be moderated by the engraftment of human CD44-derived glia. In addition, whereas the frequency of sEPSCs was significantly lower in R6/2 × rag1$^{-/-}$ striatal neurons than in rag1$^{-/-}$ WT controls, the sEPSC frequency of CD44-engrafted R6/2s was restored to levels not significantly different from those WT controls (Fig. 7d,e and Supplementary Table 8B). The apparent

significant deficit in motor performance relative to untreated controls (80%, $P<0.0001$), and that neonatal chimerization with normal hGPCs was associated with significant phenotypic rescue (60%, $P=0.038$; Supplementary Fig. 5). Analysis of top features showed no significant differences at 8 weeks between sham-treated WT and R6/2 mice in speed, stride length or duration of stride or swing. Yet by 11 weeks, while average speed remained constant across groups, the sham-treated R6/2 mice showed significant deficits in stride length ($P<0.0001$) and duration ($P=0.037$), as well as in the duration of swing phase ($P=0.001$), which were significantly, though incompletely, corrected by hGPC treatment ($P<0.0001$, $<0.01$ and $<0.0001$,

change in presynaptic inputs to R6/2 neurons likely reflected decreased release probability, since the frequency of miniature EPSCs also exhibited a trend towards disease-associated reduction and graft-associated recovery (Fig. 7d,e). While the EPSC amplitude of R6/2 striatal neurons was unaffected by chimerization (Fig. 7f and Supplementary Table 8C), the lower frequency of sEPSCs in the R6/2 MSNs, and their partial restoration by engrafted normal glia, was consistent across the spectrum of EPSC amplitudes (Fig. 7g). Of note, engraftment with normal CD44-defined glia had no effect on any electrophysiological measure in otherwise normal rag1$^{-/-}$ WT mice; only in R6/2 mice did glial engraftment affect input resistance and sEPSC frequency.

**Normal glia reduced interstitial K$^+$ in the HD striatum**. A number of studies have implicated dysfunction of neuronal potassium channels in the HD striatum[23,24], and the contribution of defective glial potassium uptake to HD pathogenesis[6]. Astrocytes play an important role in buffering K$^+$ released during synaptic transmission[25–27]. If astrocytic K$^+$ uptake is impaired, then interstitial K$^+$ rises, and the transmembrane gradient for K$^+$ is decreased, resulting in the relative

depolarization, and hence increased excitability, of local neurons[25,28]. The increased membrane resistance of WT MSNs in mHtt (73Q) glial chimeras suggested precisely such a defect in potassium handling by mHtt-expressing human glia (Fig. 3d–f). On that basis, we next used potassium microelectrodes to ask whether the hyperexcitability and increased membrane resistance of R6/2 striatal neurons was associated with elevated interstitial K$^+$ *in vivo*. We found that the levels of interstitial K$^+$ were indeed significantly higher in R6/2 mice than in WT littermates, when assessed at 16 weeks of age (Fig. 8): Whereas WT mice maintained extracellular striatal K$^+$ at a level of $3.13 \pm 0.08$ mM (mean $\pm$ s.e.m.; $n = 7$ mice), R6/2 striatal K$^+$ levels were significantly higher, averaging $3.77 \pm 0.04$ mM ($n = 8$; $P < 0.0001$, ANOVA with *post hoc* Tukey's tests; Supplementary Table 9).

We next asked whether the disease-associated elevation in extracellular K$^+$ might then be attenuated by colonization with engrafted normal glial cells, and whether that might account for the partial restoration of normal membrane resistance and firing thresholds observed in R6/2 mice transplanted with normal glia. This indeed proved to be the case, in that the R6/2 striata neonatally engrafted with normal CD44$^+$ hGPCs manifested significantly and substantially lower levels of interstitial K$^+$ ($3.31 \pm 0.10$ mM; $n = 7$ mice) than their unengrafted R6/2 littermates ($3.77 \pm 0.04$ mM, $n = 8$; as noted above) ($P = 0.0007$; Supplementary Table 9). The reduction in K$^+$ afforded by glial chimerization occurred only in R6/2 striata; WT mice transplanted with CD44$^+$ GPCs manifested extracellular K$^+$ levels no different than their untransplanted littermates ($3.32 \pm 0.06$ versus $3.13 \pm 0.08$ mM, respectively) (Fig. 8).

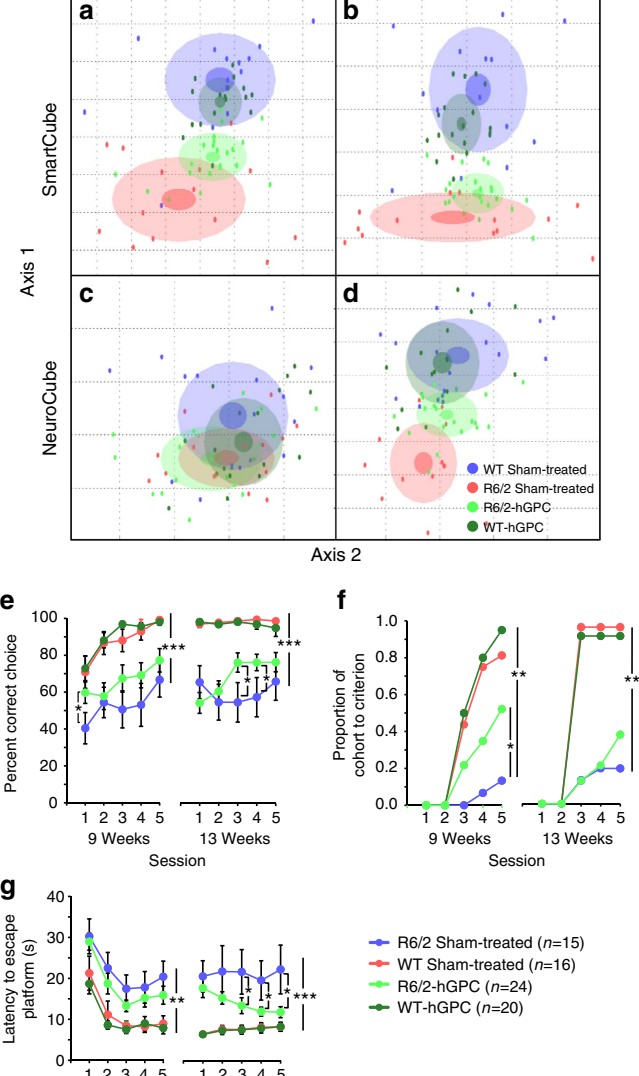

**Figure 6 | Treatment with hGPCs partially rescues disease signatures in SmartCube and NeuroCube and improved cognitive deficit in T-maze.** To build a two-dimensional representation of the multidimensional space in which the groups are best separated, we first find statistically independent combinations of the original features, pick the two new composite features (axes 1 and 2) that best discriminate between the groups, and used them as *x* and *y* axes. Each dot represents a mouse. The centre, small and large ellipses are the mean, s.e. and s.d. of the composite features for each group. The distance and overlap between the groups are used to calculate the discrimination index, which indicates how reliably a classifier can be trained to discriminate between the two groups. (**a**) SmartCube showed a significant difference between sham-treated WT and R6/2 mice (91%) at 8 weeks of age. There was a significant functional preservation (36%) in hGPC-treated versus untreated R6/2 mice. (**b**) At 11 weeks of age there was also a significant R6/2-associated deficit (92%), with a marginally significant recovery (19%) in response to hGPC treatment. (**c**) In NeuroCube, R6/2 mice had a marginally significant deficit (66%) and therefore no significant recovery by hGPC treatment could be measured. (**d**) However, at 11 weeks of age, a significant deficit (88%) and recovery (46%) by hGPC treatment was noted. The improvement of motor and cognitive behaviour of hGPC-engrafted R6/2 mice was also manifest in the T-maze test. (**e**) R6/2 mice chose the correct arm fewer times than WT mice over training at 8 weeks of age and again when retrained at 13 weeks of age (as compared with the sham-treated WT mice). GPC-treated R6/2 mice showed better performance than sham-treated R6/2 mice in specific sessions at both ages. (**f**) Fewer R6/2 than WT mice reached criterion (6 out of 8 correct trials per day for 3 consecutive days) during training and again during retraining at the older age. hGPC treatment improved acquisition in R6/2 mice during the initial training phase. (**g**) R6/2 mice were slower than WT mice to reach the platform during training and retraining. hGPC treatment improved performance during retraining. Asterisks denote significant main effects or *post hocs* and (means $\pm$ s.e.m.; *$P < 0.05$; **$P < 0.01$; ***$P < 0.001$.)

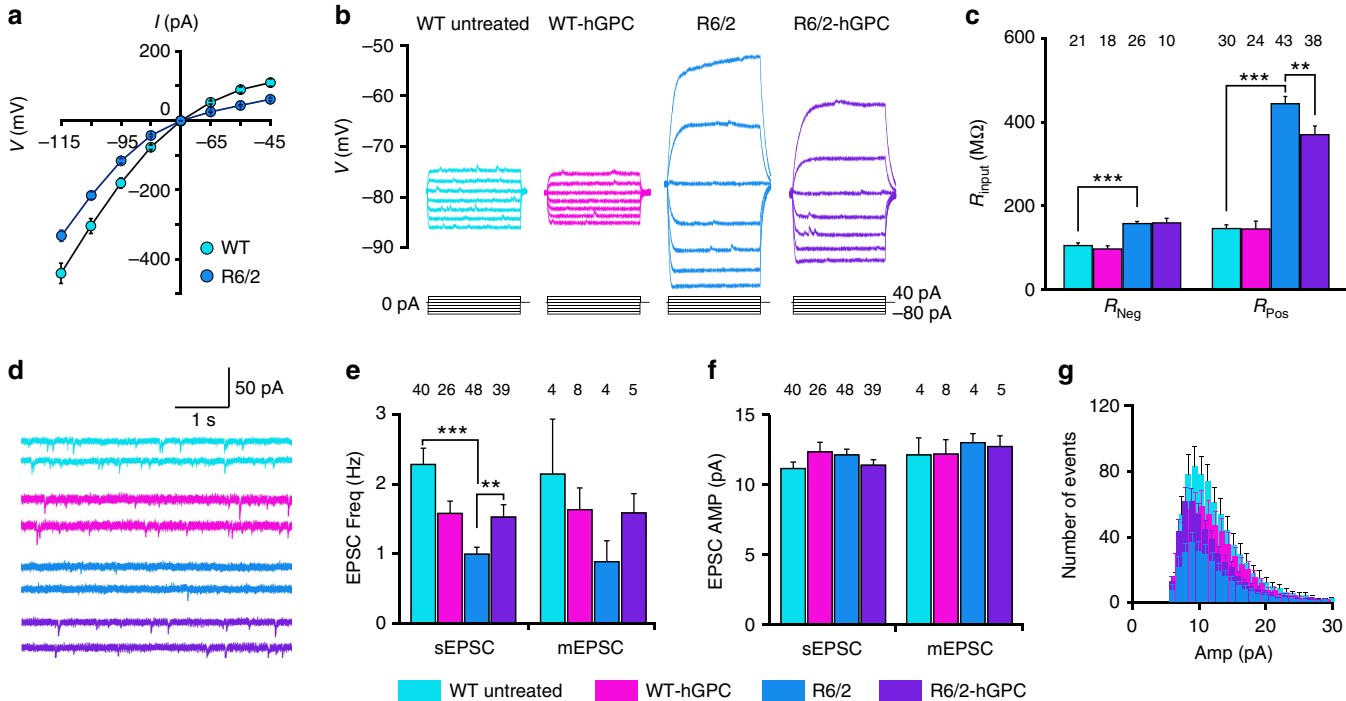

**Figure 7 | Chimerization with normal glia partially normalizes MSN physiological function.** (**a**) The current–voltage relationship (*I–V*) derived from whole-cell *V*-clamp recordings in WT and R6/2 × rag1[−/−] (R6/2) mice. (**b**) Representative whole-cell *I*-clamp recordings from rag1[−/−] WT, CD44 hGPC-engrafted rag1[−/−] WT's, R6/2 × rag1[−/−] mice and CD44-engrafted rag1[−/−] mice. Lines below each group of traces indicate the current injection steps. (**c**) The input resistance $R_{input}$ was significantly higher in R6/2 × rag1[−/−] striatal neurons than in WT × rag1[−/−] controls, but was partially restored to normal in R6/2 mice chimerized with normal CD44-sorted hGPCs. (**d**) Representative traces of sEPSCs from striatal neurons recorded in rag1[−/−] control (black), CD44-engrafted rag1[−/−] (yellow), R6/2 × rag1[−/−] (purple) and CD44-engrafted R6/2 × rag1[−/−] (green) mice. (**e,f**) The frequency (**e**) and amplitude (**f**) of sEPSCs and miniature EPSCs (mEPSCs). (**e**) The sEPSC frequency was significantly lower in R6/2 striatal neurons than in WT rag1[−/−] controls, but was restored in CD44-engrafted R6/2s to levels not significantly different from control. (**f**) In contrast, the EPSP amplitude of R/2 striatal neurons was unaffected by chimerization. (**g**) Cumulative distribution of sEPSCs. The lower frequency of sEPSCs in the R6/2 MSNs, and partial restoration by hGPC engraftment, was consistent across EPSC amplitudes. WT-untreated, $n = 11$; WT-hGPC, $n = 8$; R6/2-untreated, $n = 11$; R6/2-hGPC, $n = 8$. Means ± s.e.m.; *, ** and *** indicates $P < 0.05$, 0.01 and 0.001, respectively, by one-way ANOVA with *post hoc t*-tests.

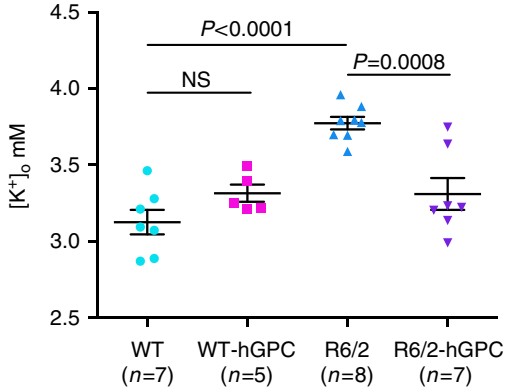

**Figure 8 | Normal glial engraftment reduces interstitial K[+] levels in the R6/2 striatum.** Potassium electrodes were used to measure the interstitial levels of striatal K[+] in both WT mice and their R6/2 littermates at 16 weeks of age (± 4 days), with and without neonatal intrastriatal transplants of CD44-sorted hGPCs. Untreated R6/2 mice manifested significantly higher levels of interstitial K, which were restored to normal in R6/2 mice neonatally engrafted with hGPCs ($P < 0.01$ by one-way ANOVA). In contrast, hGPC engraftment did not influence the interstitial K[+] levels of WT mice. All values graphed as means ± s.e.m.

## Discussion

A number of recent reports have highlighted the contribution of glial cells to the pathogenesis of neurodegenerative disorders, most particularly in the spinal cord, in which glial pathology has been implicated in the course of amyotrophic lateral sclerosis[1,2,4,5]. In this study, we asked whether glia might contribute to the genesis and progression of HD, a protoypic neurodegeneterative disorder of the brain. We found that mice whose striata were engrafted with GPCs derived from mHtt-expressing hES cells (48Q) manifested significantly slowed motor learning than littermates chimerized with normal (18Q) GPCs derived from an unaffected sibling (Fig. 2). Using mice chimerized with human fetal striatal tissue-derived glia (73Q), we then found that MSNs resident in that HD glial environment were more excitable than those engrafted with control glia (23Q), and manifested neurophysiological abnormalities previously noted in HD MSNs, both *in vitro*[7] and within the striata of R6/2 HD mice[24] (Fig. 3). On the basis of this glial-mediated recapitulation of HD-associated striatal dysfunction, we then asked whether the introduction of normal, healthy glia into the HD environment might slow disease progression. We found this to be the case, in that striatal chimerization of R6/2 mice (120Q) with normal fetal human glial cells was associated with significantly increased survival (Fig. 5), a slower rate of motor deterioration, partial rescue of behavioural abnormalities (Fig. 6 and Supplementary Figs 4 and 5), partial

normalization of MSN physiology (Fig. 7) and significant restoration of local K$^+$ homeostasis (Fig. 8). Together, these observations implicate glial pathology in the pathogenesis and progression of HD, and strongly suggest that colonization of diseased striata with healthy glia may be a viable strategy for slowing disease progression in HD.

Previous studies have highlighted the abnormal physiology of R6/2 mouse MSNs, which are characterized by a relatively depolarized resting membrane potential, increased input resistance and an increased threshold for excitatory postsynaptic potentials (EPSPs)[24]. We found that the MSNs of mice neonatally chimerized with mHtt-transduced human glia (73Q) exhibited features of these same abnormalities, and had response characteristics remarkably similar to those previously reported for R6/2 MSNs[24]. As a corollary then to these toxic effects of mHtt glia on normal MSNs, we postulated that WT glia might be capable of rescuing pathology in R6/2 neurons. This prediction that was borne out with the increased survival of R6/2 mice engrafted with WT hGPCs, as well as by the improved physiology of R6/2 neurons in the WT glial striatal environment. In particular, while striatal neurons in R6/2 × rag1$^{-/-}$ immunodeficient mice manifested the expected high input resistances and low sEPSP frequencies of R6/2 mice, those engrafted with normal CD44-sorted glia exhibited a significant reduction in input resistance and a significant increase in EPSP frequency, to levels not significantly different from WT controls (Fig. 7a,b). Notably though, whether the restoration of these physiological parameters, and more broadly the improved motor performance and survival increments associated with WT glial engraftment, is due to donor-derived astroglia or persistent glial progenitor cells remains unclear, since CD44 isolates both phenotypes, and the resultant glial chimeras are heavily colonized with each. That said, the support of striatal function afforded by CD44$^+$ astroglia and their progenitors was consistent and substantial.

A number of previous studies have reported the hyperexcitability of MSNs in HD, and several have pointed to defects in potassium conductance and potassium channel expression as contributing to both the hyperexcitability and increased input resistance of HD striatal neurons[23,24]. Among other functions, astroglia are tasked with the uptake of K$^+$ from the brain's interstitial and synaptic spaces, into which K$^+$ flows in the setting of neuronal depolarization[28,29]. If K$^+$ uptake is impaired, then the transmembrane gradient for K$^+$ is decreased, resulting in the increased excitability of local striatal neurons. Khakh and colleagues ascribed these findings to defects in astrocytic Kir 4.1 channel expression in HD mice[6], while Levine and colleagues have highlighted the contribution of neuronal downregulation of Kir2.1, Kir2.3 and Kv2.1 to the hyperexcitability of MSNs[23]. It seems unlikely that the loss of any one of these channels would be sufficient to produce HD pathology, as they manifest considerable functional redundancy, but the concurrent downregulation of a number of inwardly rectifying K$^+$ channels across both neurons and glia might be expected to exert significant pressure on striatal K$^+$ homeostasis. As such, our observation of hyperexcitability by normal MSNs resident in an HD glial chimeric environment suggests that the pathological activation patterns of R6/2 MSNs might be substantially non-cell autonomous, and elicitable in otherwise normal neurons when those neurons are faced with defective local glial potassium uptake. Indeed, it was the apparent dependence of MSN hyperexcitability on glial K$^+$ dysregulation[6], as well as the coincident observation that HD-related muscle hyperexcitability similarly reflects poor K$^+$ conductance[30], which suggested that local K$^+$ gradients, and thus MSN firing thresholds, might be restored by colonization with WT glia.

Together, these data suggest a significant role for glial dysfunction in HD pathogenesis, and hence the potential value of glial replacement in HD therapeutics. The mHtt human glial chimeric mouse model established here permitted us to evaluate the mechanisms of neurotoxicity of mHtt-expressing glia on normal neostriatal neurons. Furthermore, our model allowed us to isolate and investigate the non-cell autonomous component of HD neuropathology, and to do so in a humanized glial context, as opposed to the transgenic rodent context in which artefactually longer polyglutamine expansions are expressed by neurons and glia alike. Using this strategy, we established that a significant degree of protection may be offered to vulnerable mHtt-expressing neurons by an improved striatal glial environment, strongly suggesting the therapeutic potential of a glial replacement strategy in HD. Our finding that interstitial potassium levels are higher in the R6/2 striatum than in its WT counterpart, and may be substantially normalized by chimerization with normal glia (Fig. 8), further highlights the potential of glial cell replacement as a means of ameliorating HD-related pathology. As such, our data lend strong support to the possibility of transplanting normal glial progenitor cells into the HD striatum, both as for the treatment of manifest HD, and as a means to delay disease appearance in premanifest cases. Indeed, given the success in animal models of strategies developed to trigger the production of new striatal neurons from resident neural stem cells[31], which have substantially extended the survival of R6/2 mice[17,32,33], we may postulate that the combination of induced neuronal replacement with WT glial engraftment may act synergistically to preserve function in the diseased HD striatum. In concert with complementary strategies, such as genetic correction of mHTT alleles in patient-derived induced pluripotential cells before their induction as glial progenitors[34], which might then permit the delivery of autologous glial progenitors[35], these advances may enable clinically meaningful therapeutic options for this hitherto underserved and untreatable patient population.

## Methods

**Isolation of fetal human astroglial progenitor cells.** Human fetal brain tissue was obtained from aborted fetuses (18–22 weeks gestational age), with maternal consent and under protocols approved by the University of Rochester-Strong Memorial Hospital Research Subjects Review Board. Briefly, forebrain tissue was minced and dissociated using papain and DNase as previously described[36–38], always within 2 h of extraction. The dissociated cells were maintained overnight in DMEM/F12/N1-based medium supplemented with 10 ng ml$^{-1}$ FGF2. Astrocyte-biased glial progenitor cells were isolated from the tissue dissociates using magnetic activated cell sorting targeting the astroglial hyaluronate receptor CD44 (ref. 12) using conjugated microbeads (Miltenyi) according to the manufacturer's instructions. Cytometry confirmed that >95% of cells expressed CD44 immediately after sorting. At that point, the cells were resuspended in DMEM/F12/N1 supplemented with 10 ng ml$^{-1}$ bFGF and 2% PD-FBS at 2.5 × 10$^5$ cells per ml in six-well suspension plates, in preparation for either transduction and further expansion, or for direct transplantation.

**Production of mHtt-transduced glial progenitor cells.** To express mutant versus control Htt in human GPCs, we used a self-inactivating lentiviral system[39] to over-express either mutant (73Q) or normal (23Q) Htt. To this end, we constructed a plasmid (pTANK-CMVie-Htt-IRES-LckEGFP-WPRE) to carry, in the 5′–3′ direction, the cPPT element[40]; the cytomegalovirus immediate early promoter; the expression cassette of the first exon of the huntingtin gene and membrane-bounded EGFP, expressed in tandem under the Internal Ribosome Entry Site (IRES)[41], and the Woodchuck Hepatitis Virus Posttranscriptional Regulatory Element (WPRE)[42]. The control virus expressed only LckEGFP. Virus particles pseudotyped with vesicular stomatitis virus G glycoprotein were produced, concentrated by ultracentrifugation and titrated on 293HEK cells. Following fetal cell dissociation and CD44-based immunomagnetic sorting, the cells were transduced with Lenti-htt23Q-LckEGFP, Lenti-htt73Q-LckEGFP or Lenti-LckEGFP control virus, each at 5 multiplicities of infection. Transduced cells were isolated 5 days later, following EGFP-directed FACS. The

cells were then maintained in suspension in low serum and FGF2-containing media before transplant.

**Production of GPCs from embryonic stem cells.** Glial progenitor cells were generated from hESCs using our previously described protocols[14]. Cells were collected between 160 and 240, by which time the majority typically expressed the bipotential glial progenitor cell marker CD140a, while the remainder were A2B5$^+$/CD140a$^-$ astrocytes. hESCs were obtained from GENEA, Inc. (Sydney, Australia), as lines GENEA19 (normal Htt: 18 CAG) and GENEA20 (mHtt: 48 CAG), which were derived as a sibling pair from one couple[15].

**Animals.** Two strains or mice were used in this studies: rag1$^{-/-}$ and R6/2 × rag1$^{-/-}$ The overall distribution of mice in different experimental end points is shown in Supplementary Table 1. R6/2 × rag1$^{-/-}$ mice are heterozygous transgenic R6/2 mice[13], transgenic for the 5′-end of the human Huntingtin gene and bearing a 120 ± 5 CAG repeat expansion in the first exon of the HTT gene, then bred with rag1$^{-/-}$ homozygous immunodeficient mice. The mice were bred through consecutive transplantation of R6/2 ovaries derived from R6/2 × rag1$^{-/-}$ mice into WT B6CBAF1/J females at Jackson Laboratories (Bar Harbor, ME). Genotyping was performed by PCR analysis of genomic DNA isolated from tail clippings following the Jackson Laboratories genotyping protocol. The mice were socially housed under micro-isolator conditions, with *ad lib* access to food and water. All procedures were performed in agreement with protocols approved by the University of Rochester Committee on Animal Resources.

**Cell preparation for transplantation.** CD44-sorted and stably expressing glial progenitor cells, biased to astrocyte fate[43], were passaged with TrypLE 3–5 days before transplantation into R6/2 mice. The passaged cells were plated at a density of 100,000–150,000 cells per ml into 100-mm ultra-low attachment plates in the media described above to allow small cell clusters (100–200 μm in diameter). To prepare cells for transplantation, cells were collected, spun down, washed with Ca$^{2+}$/Mg$^{2+}$-free Hanks' balanced salt solution (HBSS) and resuspended to a final concentration of 10$^5$ cells per μl in Ca$^{2+}$/Mg$^{2+}$-free Hanks' balanced salt solution.

**Transplantation.** R6/2 × rag1$^{-/-}$ and rag1$^{-/-}$ littermates were transplanted within 24 h of birth, postnatal day 1 (P1). The mice were anesthetized by deep hypothermia and transplanted either bilaterally in the striatum with a total of 100,000 cells (50,000 cells per hemisphere; two-point transplantation paradigm) as described[36]. The cell transplant procedures were conducted under aseptic conditions.

**Histology.** Animals were killed using sodium pentobarbital and perfused transcardially with saline followed by 4% paraformaldehyde, and their brains were processed for immunocytochemistry as previously described. Sagittal equidistant cryosections of sections (20 μm) spanning the whole brain were analysed. Human cells were identified through immunostaining with anti-human nuclear antigen (1:800, MAB1281, Millipore, Temecula, CA, USA). Engrafted human cells were mapped using Metamorph imaging software and an automated fluorescence microscope (Leica Microsystems, Wetzlar, Germany). Brain sections were then co-stained to define phenotype, using combinations of the following antibodies: mouse anti-huntingtin antibody clone EM48 (1:250, MAB5374, Millipore); mouse anti-glial fibrillary acidic protein (GFAP) (1:800, SMI-21, Covance, Princeton, NJ, USA); rabbit anti-Olig2 (1:400, RA25081 Neuromics, Edina, MN, USA); rabbit anti-PDGFRα (1:400, 5241 Cell Signaling Technology, Danvers, MA, USA). Slides were analysed serially every twenty-fourth section using the optical fractionator method to estimate the total number of engrafted human cells of each histological marker (GFAP, Olig2, PDGFRα) using StereoInvestigator imaging software (MicroBrightField, Burlington, VT, USA).

**Motor assessment.** Mice from different litters were randomly assigned for rotarod evaluation. The mice were handled under the same conditions by one investigator at the same day and time. The female[44] experimenter was blind as to the genotype and treatment of the mice. The mice were tested every 4 weeks, in three rotarod trials per session, allowing at least 5 min of rest between each trial. The rotarod (Ugo Basile) accelerated from 5 to 40 r.p.m. and each trial lasted 5 min. The three values were averaged, and the data were analysed using two-way ANOVA (treatment × genotype) using GraphPad Prism v.5.0b (GraphPad, San Diego, CA).

**Survival.** Mice from multiple litters were assigned to the experiment. The mice were kept in mix genotype with three to five mice per cage. To determine lifespan, the mice were checked once a day at a younger age. Diseased mice were checked twice a day. The criterion for euthanasia was determined at the point in time when R6/2 mice were found moribund and could no longer right themselves after 30 s when placed on their side. Deaths that occurred overnight were recorded the following morning.

**Statistics.** Statistical analyses and graphs were generated using GraphPad Prism v.5.0b (GraphPad Software, San Diego, USA). Results are expressed as mean ± s.e.m., except where otherwise noted. Comparisons between groups were typically performed using ANOVA with Tukey's multiple comparisons tests. Survival data were analysed by Log-rank (Mantel Cox) testing. Significance was defined as $P \leq 0.05$.

**Electrophysiological recordings.** Mice at 12 weeks after birth were deeply anaesthetized with ketamine (100 mg kg$^{-1}$) and xylazine (10 mg kg$^{-1}$). An open window on the top of the mouse skull was cut and the brain was removed rapidly into the oxygenated, ice-cold, cutting solution, then glued to the stage of a Leica VT1000S vibratome (Leica Biosystems, Buffalo Grove, IL, USA), with the posterior surfaces down. Transverse brain slices of 300 μm were cut in the oxygenated, ice-cold, cutting solution containing (in mM): 2.5 KCl; 1.25 NaH$_2$PO$_4$; 10 MgSO$_4$; 0.5 CaCl$_2$; 10 glucose; 26 NaHCO$_3$; and 230 sucrose. Slices containing the striatum were incubated in the slice solution gassed with 5% CO$_2$ and 95% O$_2$ for at least 1 h, before being placed in a recording chamber (1.5 ml), which was superfused with the slice solution gassed with 5% CO$_2$/95% O$_2$ at room temperature (23–24 °C). The slice solution (artificial cerebrospinal fluid, aCSF) contained (in mM): 126 NaCl; 26 NaHCO$_3$; 2.5 KCl; 1.25 NaH$_2$PO$_4$; 2 MgSO$_4$; 2 CaCl$_2$; 10 lactate; and 10 glucose.

The recording chamber was placed on the stage of an Olympus BX51 upright microscope (Olympus Optical Co., NY, USA) equipped with DIC optics, and cells were visualized with a × 60 water immersion lens. Patch electrodes with a resistance of 7–9 MΩ were pulled from TW150F-4 glass capillaries (i.d. 1.12 mm, o.d. 1.5 mm, World Precision Instruments, USA) using a PC-10 electrode puller (Narishige International USA, Inc. East Meadow, NY, USA). For normal hGPC-engrafted R6/2 mice, MSNs were identified morphologically under DIC optics in randomly chosen striatal fields, while in Q23 versus Q73 mHtt$^+$ hGPC-chimeric striata, MSNs were identified in the vicinity of mHtt:EGFP-expressing glia under two-photon microscopy. In both cases all neurons were patched with the patch pipette filled with the pipette solution (mM): 140 potassium gluconate; 2 MgCl$_2$; 10 HEPES; 4 Mg-ATP; 0.3 Na-GTP; and 5 sodium phosphocreatine (pH 7.3). A seal resistance < 5 GΩ was rejected.

Membrane currents and potentials were recorded under the voltage-clamp and current-clamp configurations, respectively, with Axopatch MultiClamp (Axon Instruments, Forster City, CA, USA), interfaced to a desktop IBM-compatible computer via a Digidata A/D 1440A converter digitizer (Axon Instruments, Forster City, CA). Recording signals were filtered through a low-pass filter with a 2-kHz cut-off frequency and sampled by pCLAMP 10.2 software (Axon Instruments) with an interval of 50 μs.

**Interstitial potassium recordings.** *In vivo* recordings were obtained from the cortex and striatum of 12- to 18-week-old R6/2 HD mice and WT littermates, as well as from their matched littermates transplanted neonatally with a total of 2 × 10$^5$ CD44-sorted hGPCs, with bilateral injections of 5 × 10$^4$ cells each into striatum and the parietal cortical mantle. Mice were anesthetized using isoflurane (1.5% mixed with 1–2 l min$^{-1}$ O$_2$), and their heads restrained with a custom mini-frame, to which the mice were habituated the day before in multiple sessions, with a total training duration of 2 h. On the day of recording, a 3-mm craniotomy was opened over the motor cortex (centred at 1.5 mm lateral to bregma), and the dura removed. The procedure lasted < 20 min to minimize anaesthesia exposure during recording, and 30 min recovery from isoflurane anaesthesia was allowed post-operatively. Body temperature was maintained throughout with a heating pad. The aCSF solution contained (in mM) 150 NaCl, 2.5 KCl, 1.25 NaH$_2$PO$_4$, 2 MgCl$_2$, 2 CaCl$_2$, 10 glucose and 26 NaHCO$_3$, pH 7.4.

Ion-sensitive microelectrodes for K$^+$ recording were prepared as glass pipettes (WPI, TW150-4) with a tip diameter of 1–3 μm, as previously described[28,29]. The pipettes were silanized with dimethylsilane I (Fluka, Sigma) and filled with K$^+$ ionophore I cocktail B (Fluka). The backfill solution for K$^+$ ion-specific meter (Hanna Instr., RI) was 0.15 M KCl. K$^+$ electrodes were calibrated before and after each experiment and the calibration data were fitted to the Nikolsky equation to determine electrode slope and interference[28,29]. The K$^+$ electrodes were inserted vertically into the brain within the craniotomy site, at 2.0 mm lateral to bregma, at depths of 600 μm and 2.2 mm for the cortex and striatum, respectively. Mean values over 20–30 min recording periods were utilized.

**Behavioural analysis.** Behavioural analysis was done at Psychogenics, Inc., in Tarrytown, NY. A total of 71 mice were used in Smartcube and Neurocube behavioural experiments, and 64 were used in T-maze experiments. Psychogenics staff were blinded as to whether the tested animals were transplanted or controls, and the group identities of analysed animals were not revealed to Psychogenics until after data analysis was complete and all data had been provided to the investigators. By this means, we sought to validate, through an independent blinded replication, our conclusion that normal glial implantation is sufficient to improve disease phenotype in HD.

PsychoGenics' high-throughput systems for behavioural analysis used in this study included SmartCube and NeuroCube. These capture different domains of behaviour that include cognitive, motor, circadian, socialization, gait and anxiety-state end points, among others, using custom-built computer vision software and

machine learning algorithms[21]. Experiments were conducted using modified Intellicage units (www.tse-systems.com), each with a camera mounted on top of the cage for computer vision analysis. These cages have four corners with small doors, containing antennas to pick up each animal's ID from their implanted chips, while providing access to water bottles and allowing measurement of nose-poking and cognitive performance[45,46].

The NeuroCube platform uses computer vision to detect changes in gait geometry and dynamics. Mice are placed in the Neurocube system and allowed to explore freely for 5 min while a camera placed below the transparent floor recorded their movement for automated analysis of gait and paw placement. In this study, mice were tested at 8 and 11 weeks of age. SmartCube similarly uses computer vision and mechanical actuators to detect changes in body geometry, posture and spontaneous behaviour, as well as reactions to particular challenges[19]. Throughout 45-min testing sessions, mice were alternatively experienced periods of free exploration, motor challenges, inescapable electric shock and startle-evoking stimuli. For the latter, nozzles in the wall of the chamber were used to produce brief, intense puffs of air to elicit startle responses; in some trials, these were preceded by soft (20 dB above background noise) acoustic stimuli to elicit prepulse inhibition of startle. On testing days mice were brought to experimental rooms and allowed to acclimate for at least 30 min before testing. Mice were tested at 8 and 11 weeks of age.

The T-maze is an aquatic maze that mice navigate to reach a submerged (0.5 cm below water level) escape platform. The maze consists of two enclosed waterways that meet at a perpendicular junction, forming a 'T'-shape. The walls of the maze are opaque, and the water ($25 \pm 1\,°C$) is coloured to eliminate external cues. Training for this task challenges mice to learn and recall the location of the submerged escape platform without access to external cues. Mice were tested in the procedural T-Maze at 9 and 13 weeks of age, with each week of testing consisting of 5 consecutive days of training, and each day including eight trials for every mouse. A trial began when mice were lowered on an automated platform into the water at the distal 'stem' of the 'T', and allowed to swim freely. At the junction of the perpendicular arms of the maze, the mice must decide to swim in one direction or the other, but only one path lead to a submerged escape platform. The location of the platform was kept constant for each mouse, and with successive training mice learned to choose the path containing the escape platform over the other path. Behavioural decisions in the T-maze were defined by movement across the threshold of a given path, and were coded 'right' or 'wrong choice' relative to the location of the submerged platform. Mice that failed to leave the stem of the T-maze were coded as 'no-choice'. Mice that failed to reach the platform, whether due to 'no-choice' or 'wrong-choice' behaviours, were placed directly on the platform after 60 s.

Statistical analysis of our behavioural data considered Genotype, Treatment, and as appropriate, a repeated-measures factor. For analyses in Smartcube and Neurocube (Fig. 6 and Supplementary Figs 4 and 5), the repeated-measures factor was Age (8 weeks versus 11 weeks). For analysis of T-maze data, the repeated measures factor was Session (1–5), and separate analyses were conducted at 9 and 13 weeks. Significant ($P < 0.05$) main effects and interactions in two- or three-way mixed models (SAS 9.4) were followed by post hoc comparisons ($t$-tests for planned comparisons). We only investigated the following comparisons: sham-treated WT mice versus hGPC-treated WT or sham-treated R6/2 mice; and hGPC-treated versus sham-treated R6/2 mice. Data are represented as the mean and s.e.m.

Principal component analytics were then used to assess overall group-specific treatment effects. To reduce the effect of the correlations among the many behavioural features collected we form statistically independent combinations (aka decorrelated features) of the original features, in this way also reducing dimensionality. We then apply a proprietary feature ranking algorithm based on the feature's discrimination power (that is, ability to separate groups). This ranking is used to weigh each feature by its relevance for the final step of classification. We thus do not resort to conventional feature selection, but rather weight features to diminish the effect of irrelevant information contained in lesser features. In the new decorrelated feature space, the distance and overlap between Gaussian distributions representing each group (aka clouds), are used to calculate a quantitative measure of separability (with 50% representing chance and 100% perfect separability for equally sized groups). To estimate how likely it is that such separation is simply due to chance, the obtained classifier is challenged many times with correctly labelled samples (for example, WT or treated), or with randomized labels. The overlap between the distributions of discrimination values obtained for these two labelling categories represents the probability of finding the observed discrimination by chance. For visualization, we plotted each cloud on the two most important axes (the two composite features that best discriminate between the groups), with its semi-axes equal to one standard deviation of the group. To analyze cell treatment effects, we first trained with the sham-treated WT and R6/2 mice and then asked the trained algorithm to classify the hGPC-treated mice. To interpret the effects we decomposed the trajectory of the treated groups into two vectors: one along the direction of the disease signature (the line connecting the centres of the sham-treated WT and R6/2 clouds which captures disease-specific effects) and an orthogonal component (disease unspecific effects). Rescue or recovery is given by the movement of the hGPC-treated R6/2 group toward the sham-treated WT group on the disease line. Unspecific effects can be measured then by movement of this group in the orthogonal direction, and by movement of the hGPC-treated WT group in any direction.

**Data availability.** All relevant data and methodological detail pertaining to this study are available to any interested researchers upon request to Dr Goldman.

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

## Acknowledgements

This study was supported by CHDI, NIH R01NS75345, the Leila Y. and G. Harold Mathers Charitable Foundation, the New York State Stem Cell Research Program (NYSTEM), and the Dr Miriam and Sheldon G. Adelson Medical Research Foundation. We thank Iben Lundgaard, Josh Geiger and Hongymi Kang for advice and assistance; and Vahri Beaumont for valuable comments on our electrophysiological data.

## Author contributions

A.B. and S.G. coordinated the overall study and analysed the primary data; X.L. and D.C.M. prepared the cells for the study, under S.W.'s direction; A.B. and J.M. established the glial chimeras, with M.W.; S.H., J.M., H.B. and M.T. did the histology and behavioural analyses; M.T. was responsible for genotyping and husbandry, as well as survival analysis; Q.J.X., N.K. and F.W. did the electrophysiology experiments; F.W., J.K. and M.N. analysed the electrophysiology data, with input from M.O.; Q.J.X. and F.D. performed the interstitial potassium measurements; M.N. coordinated the electrophysiology and potassium level experiments; D.B. and P.C. respectively designed and analyzed the behavioral experiments at Psychogenics; A.B., I.M.S., M.N. and S.G. defined the experimental priorities, and S.G. wrote the paper.

## Additional information

**Competing financial interests**: Drs Goldman and Windrem hold a patent on human glial chimeric mice, US 7,524,491, from which they receive no financial remuneration. None of the authors have relevant financial sakes in this work. Drs Curtin and Brunner are employees of Psychogenics, Inc.

**How to cite this article**: Benraiss, A. *et al.* Human glia can both induce and rescue aspects of disease phenotype in Huntington disease. *Nat. Commun.* 7:11758 doi: 10.1038/ncomms11758 (2016).

