## [Peer Review File · Nature Communications]

Steven A. Goldman, M.D., Ph.D.

URMC Distinguished Professor of Neuroscience and Neurology
Dean Zutes Chair in Biology of the Aging Brain
Co-Director, Center for Translational Neuromedicine

Re: Human glia can both induce and rescue aspects of disease phenotype in Huntington Disease (NCOMMS 16-01163B)

Our response to the referees:

Reviewer #1 (Remarks to the Author):

Major claims of the paper:

To define the functional role of glia in Huntington disease, authors apply a unique experimental strategy based on the development of a human HD glial chimeric mice. The effect of huntingtin gene mutation in glial on the motor performance and neuronal activity was assessed in mice by grafting human glial cells derived from both hESC carrying huntingtin mutant or mHTT-transduced fetal forebrain glial precursor cells. To further support the role of glia in the development and progression of the HD pathology, authors engrafted healthy human glia into R2/6 HD mice partially rescuing the motor performance and neuronal function, and extending the survival of the HD mouse model.

Novelty of the paper:

The paper is very novel. This design and the clear experiments in addressing the role of glial in HD or other disease model are very unique, highly advancing our understanding of the degenerative nature of HD.

Overall quality of the data:

The data is of high quality, with excellent study designs. I only have few minor comments to improve the final version of the manuscript.

Minor comments:

1. Figure 1B shows more expression of the hNT in the cortex, then the striatum of the mouse.

The appearance of greater cortical engraftment was an artifact of uneven excitation; we have replaced Figure 1B with a better light-balanced image. We also added two higher-magnification insets to Figure 1 that show the analogous densities of human donor cell nuclear antigen staining in the cortex and striatum. Moreover, we have quantified the densities of donor cell engraftment in Table 1, which shows that the densities of cortical and striatal donor cells did not significantly differ from one another.

Reviewer responses: The image is now improved. Unfortunately, I am unable to find the quantified densities of the cortical donor cell counts, as the Table 1 only shows total and striatal donor cell counts. Can authors add this data?

We were referring to the striatal densities of the Genea 19 and 20-engrafted mice not differing from one another quantitatively, as shown in the last column of Table 1. The striatal and cortical densities are similarly analogous, to the extent that the cell counts in the images shown in Figure 1 were actually identical, at 112 cells each in the white and gray matter insets. I am sorry for not making this more clear in my response.

2. Figure 1 is missing the annotation of the 'D' panel

Fixed.

3. The graph layout in Figure 2 suggest that all time points are significantly different between the animal groups which in Table S1 it is shown that this is not the case.

We're sorry for the confusing presentation of the graphed data in Figure 2. The asterisks refer to the significant difference between the mHTT and normal GPC-engrafted mice, as determined by ANOVA. The graph is not intended to show treatment-associated differences at each time point. Rather, the more granular data presentation of Table S1 accomplishes this purpose, by showing the post hoc analyses of each comparison at each time point. We have modified the caption of Figure 2 to make this point more clear, and added the relevant statistics: Two-way ANOVA revealed both a significant treatment effect ($F(3, 593) = 39.6$, $***p < 0.0001$) and time effect ($F(7, 593) = 5.47$ $p < 0.0001$); mean {plus minus} SEM.

Reviewer response: The modified figure legend better explains the statistical analysis and helps the reader with interpretation of the data.

4. There is a mistake in referencing Figure 3D in the text of the results section, where it should be Figure 3E.

Fixed, thanks.

5. Please add statistical analysis or explain why in Figure 5 graph A, depicting multiple time points of analysis, there are no statistical analysis of time point 16 and 20 implicated in Table S3.

We thank the referee for pointing out this accidental omission. Table S3 has now been updated to show the data for the 16-week time point. However, no similar analysis is possible for the 20 week time point, because all of the control animals had either died or were unable to balance on the rod by then!

Reviewer response: If the numbers of animals able to perform the rotarod motor test at week 20 in the R2/6 hGPCs are indeed higher than in the R2/6 control mice, this should be stressed more directly in the results section than it is currently (for the reader).

We thank the referee for the suggestion, and have added this point to the caption of Supplementary Table 5.

6. In the results section referring to the disease-associated hyperactivity of the R2/6 mice, it is suggested that the hGPC-graft attenuates all three phenotypes of the measured behavior: sniffing/scanning, locomotion and rearing. Please make that more clear that the graft effected two of these readouts.

We thank the referee for pointing out our confusing wording, and have adjusted the figure legend and the text to clarify this point. The relevant text (page 8) now reads: ...sham-treated R6/2 mice were hyperactive compared to sham-treated WT mice, showing increased sniffing/scanning ($p = 0.0003$), locomotion ($p = 0.008$), and rearing ($p = 0.002$). Some of these changes were attenuated in hGPC-treated R6/2 mice, which exhibited less locomotion ($p < 0.0001$) and rearing ($p = 0.0003$), relative to sham-treated R6/2 mice.

In addition, to improve the flow and organization of the paper, we have moved the behavioral phenotyping principal component analysis data of Supplementary Figure to the main body of the

paper as a new Figure 6, to which we have added the T-maze data of the prior Supplementary Figure 6. At the same time, we have now moved the SmartCube data of the prior Figure 6 to the Supplement as Figure S4, adjacent to the NeuroCube data reported in Figure S5.

Reviewer response: The textual changes and the figure adjustments improve the flow of the paper.

7. In the results section referring to the age-dependent gait deficits of the R2/6 mice, stride duration is described which is not presented in Figure 5S. Additionally, described significant effect on the body movement variability is not implicated in the graph.

We have added body movement variability, which did not differ between groups, to Figure S5 as requested. A separate graph was not presented for stride duration, since its associated output measures stride length and swing duration are already plotted.

Reviewer response: In the results section authors refer to three different measures of movement with stride duration being one of the measures quantified separately: "Yet by 11 weeks, sham-treated R6/2 mice showed significant deficits in stride length ($p < 0.0001$), stride duration ($p = 0.037$), and in the duration of swing phase ($p = 0.0011$), all of which were significantly, though incompletely, corrected by hGPC treatment ($p < 0.0001$, < 0.01 , and < 0.0001 , respectively)." The given graph of average speed is not showing any significant changes at week 11 between the animals groups. The authors need to reevaluate their figures and the text to match the results (pg. 9).

We did not claim treatment-associated changes in ambulatory speed, but rather included that as a variable to show that GPC-associated phenotypic rescue in R6/2 was selective to some measures and modalities. To make this point more clear, we have rewritten the text as follows (pages 8-9):

Analysis of top features showed no significant differences at 8 weeks between sham-treated WT and R6/2 mice in speed, stride length, or duration of stride or swing. Yet by 11 weeks, while average speed remained constant across groups, the sham-treated R6/2 mice showed significant deficits in stride length ($p < 0.0001$) and duration ($p = 0.037$), as well as in the duration of swing phase ($p = 0.001$), which were significantly, though incompletely, corrected by hGPC treatment ($p < 0.0001$, < 0.01 , and < 0.0001 , respectively).

Please indicate the figure number in the results section referring to the significant effect of the hGPC transplant in body movement variability on pg. 9. "Interestingly, hGPC treatment reduced body movement variability in both WT and R6/2 groups, an HD-independent effect ($p = 0.003$)."

8. In the results section referring to the T-maze performance of the R2/6 mice, the graphs in figure 7S show time points 9 and 13 weeks, whereas the text mentions week 11.

We thank the referee for pointing out this error; the text should refer to time points at 9 and 13 weeks, and we have adjusted it accordingly (pages 8-9).

9. Please also be more clear in the text that both R2/6 mouse groups are significantly worse in their performance in all readouts at both time points when compared to the WT mice, but the R2/6 hGPC grafted mice were significantly better compared to the R2/6 mice.

We have now explicitly noted in the results (page 9, first paragraph) that R6/2 performance lags WT performance even with GPC engraftment.

10. Please fix the graph legends in figure S4.

Fixed, thanks.

11. In the discussion in the 1st paragraph delete word 'glia' after 'glia (23Q)'.
Done.

Influence of the paper on the field:
This paper has a potential to be highly influential in the field.

Statistics:

Statistics are of high quality. The only comment regards the statistical analysis done on the behavioral readouts in the R2/6 mice: the description of the statistical analysis is not clear regarding which analysis were applied on the data in figure 6 and supplementary figures 5 and 6.

We have changed the text in the Statistical and Informatics Analysis section (page 18), to further clarify the analyses applied to our SmartCube, NeuroCube, and T-maze data. The text now reads:

Behavioral analyses considered Genotype, Treatment, and, as appropriate, a repeated-measures factor. For analyses in Smartcube and Neurocube (Figures 6, S4 and S5), the repeated-measures factor was Age (8 weeks vs 11 weeks). For analysis of T-maze data, the repeated measures factor was Session (1-5), and separate analyses were conducted at 9 and 13 weeks.

Reviewer response: This helps with understanding and the interpretation of the data.

Thank you.

Reviewer #2 (Remarks to the Author):

This is a revised manuscript and the major concern that I raised has been addressed. Therefore I recommend publication.

Thank you.

Reviewer #3 (Remarks to the Author):

The revised manuscript is greatly improved. The authors have sufficiently addressed all my concerns. Actually, the response to reviewers' comments are really well written. If possible, I would suggest to add some of these content in the discussion. For example, admitting and discussing about having no control on limiting engraftment to cortical regions vs striatum. This does not hurt the significance of the present study, rather the story is more convincing, and would evoke more general discussion on how to successfully harness such powerful technique in the future.

Thank you. If there were room I'd be happy to, but given our space limitations, I think this and other related points regarding both the strengths and limitations of our model will have to await the next appropriate opportunity.

Thanks once again to you and the referees for all of your efforts on behalf of this paper.

My best wishes,

Steve Goldman, MD, PhD

REVIEWERS' COMMENTS:

Reviewer #1 (Remarks to the Author):

Major claims of the paper:

To define the functional role of glia in Huntington disease, authors apply a unique experimental strategy based on the development of a human HD glial chimeric mice. The effect of huntingtin gene mutation in glial on the motor performance and neuronal activity was assessed in mice by grafting human glial cells derived from both hESC carrying huntingtin mutant or mHTT-transduced fetal forebrain glial precursor cells. To further support the role of glia in the development and progression of the HD pathology, authors engrafted healthy human glia into R2/6 HD mice partially rescuing the motor performance and neuronal function, and extending the survival of the HD mouse model.

Novelty of the paper:

The paper is very novel. This design and the clear experiments in addressing the role of glial in HD or other disease model are very unique, highly advancing our understanding of the degenerative nature of HD.

Overall quality of the data:

The data is of high quality, with excellent study designs. I only have few minor comments to improve the final version of the manuscript.

Minor comments:

1. Figure 1B shows more expression of the hNT in the cortex, then the striatum of the mouse. The appearance of greater cortical engraftment was an artifact of uneven excitation; we have replaced Figure 1B with a better light-balanced image. We also added two higher-magnification insets to Figure 1 that show the analogous densities of human donor cell nuclear antigen staining in the cortex and striatum. Moreover, we have quantified the densities of donor cell engraftment in Table 1, which shows that the densities of cortical and striatal donor cells did not significantly differ from one another.

Reviewer responses: The image is now improved. Unfortunately, I am unable to find the quantified densities of the cortical donor cell counts, as the Table 1 only shows total and striatal donor cell counts. Can authors add this data?

2. Figure 1 is missing the annotation of the 'D' panel
Fixed.

3. The graph layout in Figure 2 suggest that all time points are significantly different between the animal groups which in Table S1 it is shown that this is not the case.

We're sorry for the confusing presentation of the graphed data in Figure 2. The asterisks refer to the significant difference between the mHTT and normal GPC-engrafted mice, as determined by ANOVA. The graph is not intended to show treatment-associated differences at each time point. Rather, the more granular data presentation of Table S1 accomplishes this purpose, by showing the post hoc analyses of each comparison at each time point. We have modified the caption of Figure 2 to make this point more clear, and added the relevant statistics: Two-way ANOVA revealed both a significant treatment effect ($F(3, 593) = 39.6, ***p < 0.0001$) and time effect ($F(7, 593) = 5.47 p < 0.0001$); mean {plus minus} SEM.

Reviewer response: The modified figure legend better explains the statistical analysis and helps the reader with interpretation of the data.

4. There is a mistake in referencing Figure 3D in the text of the results section, where it should be Figure 3E.
Fixed, thanks.

5. Please add statistical analysis or explain why in Figure 5 graph A, depicting multiple time points of analysis, there are no statistical analysis of time point 16 and 20 implicated in Table S3.

We thank the referee for pointing out this accidental omission. Table S3 has now been updated to show the data for the 16-week time point. However, no similar analysis is possible for the 20 week time point, because all of the control animals had either died or were unable to balance on the rod by then!

Reviewer response: If the numbers of animals able to perform the rotarod motor test at week 20 in the R2/6 hGPCs are indeed higher than in the R2/6 control mice, this should be stressed more directly in the results section than it is currently (for the reader).

6. In the results section referring to the disease-associated hyperactivity of the R2/6 mice, it is suggested that the hGPC-graft attenuates all three phenotypes of the measured behavior: sniffing/scanning, locomotion and rearing. Please make that more clear that the graft effected two of these readouts.

We thank the referee for pointing out our confusing wording, and have adjusted the figure legend and the text to clarify this point. The relevant text (page 8) now reads: ...sham-treated R6/2 mice were hyperactive compared to sham-treated WT mice, showing increased sniffing/scanning ($p=0.0003$), locomotion ($p=0.008$), and rearing ($p=0.002$). Some of these changes were attenuated in hGPC-treated R6/2 mice, which exhibited less locomotion ($p < 0.0001$) and rearing ($p=0.0003$), relative to sham-treated R6/2 mice.

In addition, to improve the flow and organization of the paper, we have moved the behavioral phenotyping principal component analysis data of Supplementary Figure to the main body of the paper as a new Figure 6, to which we have added the T-maze data of the prior Supplementary Figure 6. At the same time, we have now moved the SmartCube data of the prior Figure 6 to the Supplement as Figure S4, adjacent to the NeuroCube data reported in Figure S5.

Reviewer response: The textual changes and the figure adjustments improve the flow of the paper.

7. In the results section referring to the age-dependent gait deficits of the R2/6 mice, stride duration is described which is not presented in Figure 5S. Additionally, described significant effect on the body movement variability is not implicated in the graph.

We have added body movement variability, which did not differ between groups, to Figure S5 as requested. A separate graph was not presented for stride duration, since its associated output measures stride length and swing duration are already plotted.

Reviewer response: In the results section authors refer to three different measures of movement with stride duration being one of the measures quantified separately: "Yet by 11 weeks, sham-treated R6/2 mice showed significant deficits in stride length ($p < 0.0001$), stride duration ($p=0.037$), and in the duration of swing phase ($p=0.0011$), all of which were significantly, though incompletely, corrected by hGPC treatment ($p < 0.0001$, < 0.01 , and < 0.0001 , respectively)." The given graph of average speed is not showing any significant changes at week 11 between the animals groups. The authors need to reevaluate their figures and the text to match the results (pg. 9).

Please indicate the figure number in the results section referring to the significant effect of the hGPC transplant in body movement variability on pg. 9. "Interestingly, hGPC treatment reduced body movement variability in both WT and R6/2 groups, an HD-independent effect ($p=0.003$)."

8. In the results section referring to the T-maze performance of the R2/6 mice, the graphs in figure 7S show time points 9 and 13 weeks, whereas the text mentions week 11.

We thank the referee for pointing out this error; the text should refer to time points at 9 and 13 weeks, and we have adjusted it accordingly (pages 8-9).

9. Please also be more clear in the text that both R2/6 mouse groups are significantly worse in their performance in all readouts at both time points when compared to the WT mice, but the R2/6 hGPC grafted mice were significantly better compared to the R2/6 mice.

We have now explicitly noted in the results (page 9, first paragraph) that R6/2 performance lags WT performance even with GPC engraftment.

10. Please fix the graph legends in figure S4.
Fixed, thanks.

11. In the discussion in the 1st paragraph delete word 'glia' after 'glia (23Q)'.
Done.

Influence of the paper on the field:
This paper has a potential to be highly influential in the field.

Statistics:
Statistics are of high quality. The only comment regards the statistical analysis done on the behavioral readouts in the R2/6 mice: the description of the statistical analysis is not clear regarding which analysis were applied on the data in figure 6 and supplementary figures 5 and 6.

We have changed the text in the Statistical and Informatics Analysis section (page 18), to further clarify the analyses applied to our SmartCube, NeuroCube, and T-maze data. The text now reads: Behavioral analyses considered Genotype, Treatment, and, as appropriate, a repeated-measures factor. For analyses in Smartcube and Neurocube (Figures 6, S4 and S5), the repeated-measures factor was Age (8 weeks vs 11 weeks). For analysis of T-maze data, the repeated measures factor was Session (1-5), and separate analyses were conducted at 9 and 13 weeks.

Reviewer response: This helps with understanding and the interpretation of the data.

Reviewer #2 (Remarks to the Author):

This is a revised manuscript and the major concern that I raised has been addressed. Therefore I recommend publication.

Reviewer #3 (Remarks to the Author):

The revised manuscript is greatly improved. The authors have sufficiently addressed all my concerns. Actually, the response to reviewers' comments are really well written. If possible, I would suggest to add some of these content in the discussion. For example, admitting and discussing about having no control on limiting engraftment to cortical regions vs striatum. This does not hurt the significance of the present study, rather the story is more convincing, and would evoke more general discussion on how to successfully harness such powerful technique in the future.

Reviewer #1 (Remarks to the Author):

Major claims of the paper:

To define the functional role of glia in Huntington disease, the authors apply a unique experimental strategy based on the development of a human HD glial chimeric mouse. The effect of huntingtin gene mutation in glia on motor performance and neuronal activity was assessed in mice by grafting human glial cells derived from both a hESC carrying huntingtin mutant and mHTT-transduced fetal forebrain glial precursor cells. To further support the role of glia in the development and progression of the HD pathology, the authors engrafted healthy human glia into R2/6 HD mice, partially rescuing motor performance and neuronal function, and extending the survival of the HD mouse model.

Novelty of the paper:

The paper is definitely novel. These well-designed experiments for addressing the role of glia in HD or other disease models are unique, advancing our understanding of the degenerative nature of HD.

Overall quality of the data:

The data is of high quality, with appropriate study designs. There are only a few minor comments to improve the final version of the manuscript.

Minor comments:

1. Figure 1B shows more expression of the hNT in the cortex, than in the striatum of the mouse.

The appearance of greater cortical engraftment was an artifact of uneven excitation; **we have replaced Figure 1B** with a better light-balanced image. We also added two higher-magnification insets to **Figure 1** that show the analogous densities of human donor cell nuclear antigen staining in the cortex and striatum. Moreover, we have quantified the densities of donor cell engraftment in **Table 1**, which shows that the densities of cortical and striatal donor cells did not significantly differ from one another.

2. *Figure 1 is missing the annotation of the 'D' panel*

Fixed.

3. *The graph layout in Figure 2 suggests that all time points are significantly different between the animal groups, while Table S1 shows that this is not the case.*

We're sorry for the confusing presentation of the graphed data in **Figure 2**. The asterisks refer to the significant difference between the mHTT and normal GPC-engrafted mice, as determined by ANOVA. The graph is not intended to show treatment-associated differences at each time point. Rather, the more granular data presentation of **Table S1** accomplishes this purpose, by showing the post hoc analyses of each comparison at each time point. We have modified the caption of **Figure 2** to make this point more clear, and added the relevant statistics:

Two-way ANOVA revealed both a significant treatment effect ($F(3, 593) = 39.6$, $*p < 0.0001$) and time effect ($F(7, 593) = 5.47$ $p < 0.0001$); mean \pm SEM.**

4. *There is a mistake in referencing Figure 3D in the text of the results section, where it should be Figure 3E.*

Fixed, thanks.

5. *Please add statistical analysis or explain why in Figure 5 graph A, depicting multiple time points of analysis, there is no statistical analysis of time points 16 and 20 implicated in Table S3.*

We thank the referee for pointing out this accidental omission. **Table S3** has now been updated to show the data for the 16-week time point. However, no similar analysis is possible for the 20 week time point, because all of the control animals had either died or were unable to balance on the rod by then!

6. *In the results section referring to the disease-associated hyperactivity of the R2/6 mice, it is suggested that the hGPC-graft attenuates all three phenotypes of the measured behavior: sniffing/scanning, locomotion and rearing. Please clarify that the graft affected two of these readouts.*

We thank the referee for pointing out our confusing wording, and have adjusted the figure legend and the text to clarify this point. The relevant text (**page 8**) now reads:

...sham-treated R6/2 mice were hyperactive compared to sham-treated WT mice, showing increased sniffing/scanning ($p = 0.0003$), locomotion ($p = 0.008$), and rearing ($p = 0.002$). Some of these changes were attenuated in hGPC-treated R6/2 mice, which exhibited less locomotion ($p < 0.0001$) and rearing ($p = 0.0003$), relative to sham-treated R6/2 mice.

In addition, to improve the flow and organization of the paper, we have moved the behavioral phenotyping principal component analysis data of Supplementary Figure to the main body of the paper as a **new Figure 6**, to which we have added the T-maze data of the prior Supplementary Figure 6. At the same time, we have now moved the SmartCube data of the prior Figure 6 to the Supplement as **Figure S4**, adjacent to the NeuroCube data reported in **Figure S5**.

7. *In the results section referring to the age-dependent gait deficits of the R2/6 mice, stride duration is described, but not presented in Figure 5S. Additionally, the described significant effect on the body*

movement variability is not represented in the graph.

We have added body movement variability, which did not differ between groups, to **Figure S5** as requested. A separate graph was not presented for stride duration, since its associated output measures stride length and swing duration are already plotted.

8. *In the results section referring to the T-maze performance of the R2/6 mice, the graphs in figure 7S show time points 9 and 13 weeks, whereas the text mentions week 11.*

We thank the referee for pointing out this error; the text should refer to time points at 9 and 13 weeks, and we have adjusted it accordingly (**pages 8-9**).

Please also be more clear in the text that both R2/6 mouse groups are significantly worse in their performance in all readouts at both time points when compared to the WT mice, but the R2/6 hGPC grafted mice were significantly better compared to the R2/6 mice.

We have now explicitly noted in the results (**page 9**, first paragraph) that R6/2 performance lags WT performance even with GPC engraftment.

9. *Please fix the graph legends in figure S4.*

Fixed, thanks.

10. *In the discussion in the 1st paragraph, delete the word 'glia' after 'glia (23Q)'.*

Done.

Influence of the paper on the field:

This paper has a potential to be highly influential in the field.

Statistics:

Statistics are of high quality. The only comment regards the statistical analysis done on the behavioral readouts in the R2/6 mice: the description of the statistical analysis is not clear regarding which analyses were applied to the data in figure 6 and supplementary figures 5 and 6.

We have changed the text in the Statistical and Informatics Analysis section (**page 18**), to further clarify the analyses applied to our SmartCube, NeuroCube, and T-maze data. The text now reads:

Behavioral analyses considered Genotype, Treatment, and, as appropriate, a repeated-measures factor. For analyses in Smartcube and Neurocube (Figures 6, S4 and S5), the repeated-measures factor was Age (8 weeks vs 11 weeks). For analysis of T-maze data, the repeated measures factor was Session (1-5), and separate analyses were conducted at 9 and 13 weeks.

Reviewer #2 (Remarks to the Author):

The work of Benraiss et al. addresses significant questions in stem cell biology and transplants. The work is original and of general interest. They evaluate whether transplantation of glia offers therapeutic benefit in mouse models of HD. The authors demonstrate that glia derived from human ESC cells improve rotarod, had decreased striatal atrophy and improved measures with the SmartCube. The effects on lifespan are modest. They also engrafted glia from human expressing mutant HTT (derived

from ESCs) and demonstrated significant contribution to disease pathology and behavioral changes. The strength of the investigation is the significant questions addressed. There are some points that should be further clarified.

(1) It is not clear if the same production and purification of CD-44-defined astroglial progenitors and bipotential oligodendrocyte-astrocyte was used for the experiments shown in the work. The manuscript reads as if they were distinct. If each preparation was done differently than it may not be that glia positive for expansion promotes disease and those negative are neuroprotective. It could be how the cells were made. This needs further clarification. Use of GENE20 and GENE 19 would make the studies consistent.

We're sorry if the rationale for these choices is unclear. The Genea19 and 20 GPCs were driven to a bipotential astrocyte-oligodendrocyte phenotype, a phenotype whose behavior we have extensively reported in the past (e.g., Sim et al., Nature Biotechnology, 2011; Wang et al., Cell Stem Cell, 2013). However, in the myelin wild-type environment of the normal mouse brain, as well as in R6/2, these cells differentiate as astrocytes, with essentially no oligodendrocytic differentiation, traversing a CD44 positive stage during that process. Since it became apparent that astrocytes were the phenotype responsible for the Genea20-associated pathology, we used CD44-derived cells, which largely overlap and derive from the bipotential phenotype and comprise a more astrocyte-biased pool, for the rescue experiments in which donor cells were injected into R6/2 neonates. These are simply adjacent, and largely overlapping, stages in the same lineage; our use of CD44-sorting to isolate a homogeneous pool for the rescue experiments was intended to build upon the information gained from the Genea19 vs 20 hGPC transplants, while providing a well-characterized phenotype for future experiments intended to rescue HD phenotype. As to the source of CD44 hGPCs, their acquisition from fetal tissue provided us a benchmark population, unencumbered by criticisms that might be leveled at the use of use one hESC line or another.

(2) Why is the electrophysiology done in Figure 3A-B with a lentivirus mHTT transduced system? Was it not sufficient to transplant the CD-44 positive astrocytes from the GENE20 with 48 repeats and obtain alterations in electrophysiology? Why not report this if that is the cause. The shift in systems and rationale seem out of place.

We used the lenti-mHTT transduced cells so as to introduce a longer CAG repeat expansion (73Q) than available from hESC systems, all of which have polyglutamine expansions in the 40-50Q range. We did so to accelerate the development of pathology, so as to maximize our ability to detect glial mHTT-dependent neuronal pathology. This is described on **page 5**:

To better understand the physiological basis for the relatively impaired motor performance of mHtt glial-engrafted mice, we next asked whether chimerization with mHtt glia influenced the physiology of medium spiny neurons. To that end, we established striatal glial chimeras in otherwise wild-type immunodeficient mice, via neonatal intrastriatal injection of mHtt-expressing human fetal glia. For this purpose, we used mHtt-transduced fetal tissue-derived hGPCs rather than HD hESC-derived GPCs, so as to assess the effects of mHtt bearing longer CAG repeats than the 48Q mHtt expressed by GENE 20-derived hGPCs. We postulated that longer CAG repeat expansions would accelerate glial pathology, and thus potentiate detection of paracrine neuronal dysfunction at the relatively young ages and compressed experimental time frames used in this study. To that end, we isolated hGPCs from 18-20 week human fetal forebrain, using immunomagnetic sorting directed against CD44, which as noted is highly expressed by astrocyte-biased glial progenitor cells¹². We then transduced these cells with a lentiviral vector encoding the first exon of the HTT gene bearing either mutant (73Q) or

normal (23Q) huntingtin, each upstream to an EGFP reporter, and then injected the transduced cells into the striata of neonatal *rag1^{-/-}* immune deficient mice.

(3) Could the authors in the material methods present a table of mice transplanted per study and how they were used? It seems like a separate set where made to ship to Pschogenics, Inc. Did this set have survival measured as well?

Was there any effect on weight in the studies?

No. As would be expected, all R6/2 mice lost significant weight as function of age ($F(1, 36)=8.40$, $P=0.006$ by 2-way ANOVA). But striatal hGPC engraftment did not further affect the weight of these R6/2 mice, relative to untreated R6/2s, at either 8 or 16 weeks ($F(1, 36)=0.1246$; $p=0.73$ by 2-way ANOVA). We have added these data to the text (**page 6**), and as a new **Supplementary Table S5**.

Supplementary Table S5 (Related to Fig. 5)

Weight analysis in R6/2 as function of treatment and time (in g)

Age	R6/2-hGPC	R6/2-Untreated
8 weeks	22.3 ± 0.8 (n=12)	20.3 ± 1.7(n=11)
16 weeks	17.4 ± 0.3 (n=8)	18.6 ± 0.7 (n=9)

Mean (weeks) ± S.E.M.

Did the response occur in both females and males?

Equal numbers of females and males were assigned to all experiments. As noted (**page 7**), we did not notice any significant difference in life span between genders, in either treated or untreated R6/2 mice. To address the referee's concern, we have added a table describing the median survival of males and females, as a new **Supplementary Table S6**.

Supplementary Table S6 (Related to Fig. 5)

Analysis of survival of hGPC-engrafted R6/2 mice by gender

	Male survival	Female survival	p value (t test)
R6/2-hGPC	18.8 ± 0.7 (n=16)	19.6 ± 1 (n=13)	ns
R6/2-untreated	18.0 ± 0.8 (n=13)	16.9 ± 1 (n=15)	ns

Mean (weeks) ± S.E.M.

(4) The transplantation is done 24 hours after birth and this does not seem like it directly tests the whether these cells can be used therapeutically in HD. Do the authors have benefit if the transplantation is done when the mice are adults?

This study was done to establish the dual principles that mutant HTT-expressing glia contribute to pathogenesis and disease progression in HD, and that their replacement by normal glia might thereby delay disease progression. Our experiments now strongly and unequivocally support those hypotheses, and provide a basis for further studies across the age spectrum to better define the age range and stages of disease progression at which glial replacement might be effective, and at what point in adult disease progression such a strategy might become ineffective. This issue is not as simple as transplanting adult R6/2s, since the age of the animal, its CAG repeat length and extent of disease progression at that age will influence therapeutic efficacy. Such preclinical modeling will likely require

several years of serial studies, which we trust the referee will agree are beyond the scope of our present study.

(5) *Did two separate studies of the transplanted cells into mice give similar trends?*

Yes; the survival and behavioral analysis groups were established and run separately, at different times of the year by different technicians using different cells, as were the animals subjected to behavioral analysis at Psychogenics. The beneficial effect of glial progenitor transplantation and glial replacement in these HD mice is both robust and reproducible.

(6) *Is Table 1 described in the results section?*

It is now, in separate references to **Table 1A (page 4)** and **Table 1B (page 6)**.

(7) *Figure 2 does give a sense of the variability in rotarod for the different mice as plotted.*

We have expanded the caption to Figure 2 to improve the clarity of its statistical analysis.

(8) *Some figures have the number of animals used while others do not. Perhaps specify in each figure. This has been corrected throughout the text and figures.*

Reviewer #3 (Remarks to the Author):

Comments:

In this manuscript, the authors try to examine the role of glia in Huntington's disease (HD). In a previous study, the authors established a protocol to generate Glial Progenitor cells (GPCs) and their derived astroglia and oligodendrocytes from human embryonic stem cells (hESCs). In this work, the authors try to explore whether neonatal engraftment of these cells into immunodeficient mice could have therapeutic effect in mouse models of HD. Stem cell therapy has big potential for treating neurodegenerative diseases. Previous studies have attempted to use derived neuron engraftment for treating HD. In this study, the authors use a chimeric mouse where control and mutant GPCs generated from human embryonic stem cells are engrafted in the striata of immunodeficient or R6/2 mice. Using a combined behavioral and electrophysiological approach, the authors suggest that glial pathology contributes to HD by increasing interstitial potassium in the striatum. The mechanism still remains unknown. But the authors showed that glial regulation of extracellular potassium level is involved. Overall, the observation is interesting. The current study seems to be preliminary. A few important issues need to be addressed.

Major issues:

1. *"data not shown" appears several places. In general, this statement does not help. The authors should either show the data, or eliminate such statement throughout the manuscript.*

We have removed the 2 references to data not shown, which were both on page 4. In the first instance, the data were actually stated as such – that no HuD/C cells or residual pluripotential gene expression was noted in the Genea ESC-derived GPCs in vitro. In the second instance, the lack of in vivo neuronal production or tumor formation were left described as such, and the reference to data not shown similarly removed. In each of these instances, negative data were reported, for which there was frankly nothing to show.

2. *The extend of GPCs engraftment is not well-characterized. The authors focus on the striatum, which*

is the primary region for neurodegeneration in HD. However, in their studies, the cortical regions are also largely affected. It is known that pathophysiological changes in cortex also contribute to HD. Therefore, it is critical to perform controls. GPCs engraftment should be well controlled in striatum alone, cortex alone, etc. In addition, the authors should also take into consideration of the extend of engraftment, region, cell counts, etc. It is critical to perform experiments with high rigor in order to reach conclusion about causal effect.

We agree with the referee as to the need for rigorous evaluation of cell donor cell engraftment, and have provided extensive quantification and imaging of engraftment in our animals. However, we have no control on limiting engraftment to cortical regions vs. striatum. The cells are highly migratory, and traverse compartments with ease, as a function of time and initial injection site. Thus, the requested experiments are simply not biologically possible at present, at least with our current level of understanding.

3. Fig 1A: it would be more ideal to compare multiple consecutive sections taken at similar locations (the olfactory bulb is present in some sections but not others). The way it is currently presented implicates differences in spread between the experimental groups. Does the extend of engraftment correlate with the behavioral outcome?

We have corrected **Figure 1A** so that the dot-map schematics of the control and experimental-implanted animals are shown at precisely identical mediolateral sagittal planes. The level of engraftment, as noted was indistinguishable between the groups.

d4. Tumorigenesis is generally a big potential problem for stem cell therapy. The authors did indicate that no tumor was observed in all studies. However, this should be taken into more serious consideration for in vivo studies. Tumor markers should be examined using IHC.

We do indeed take the possibility of tumorigenesis seriously. My lab was one of the first to report tumorigenesis from hESCs as a significant concern limiting therapeutic progression (Roy et al., Nature Medicine, 2006), and I've reviewed this issue several times (Goldman et al., Science, 2012; Fox et al., Science 2014; Goldman, Cell Stem Cell, 2016). Indeed, we developed the long hESC GPC induction and differentiation protocols used in this study specifically to minimize the possibility of tumorigenesis (Wang et al., Cell Stem Cell, 2013), successfully so. On a personal note, I'm trained, and remain clinically active, as a neuro-oncologist, and I look at these engrafted mouse sections as I would any patient's neuropathological samples. To my eye, none of the animals presented in this paper manifested any evidence of either teratomatous or neuroglial tumors. I should also note for the referee's interest that are no malignancy-specific tumor markers for neuroepithelial or primitive neuroectodermal tumors, which would have been the most likely tumor types of potential concern.

5. Does fetal forebrain hESC studies (hGPC-23Q and hGPC-73Q) engraftment also cause similar decline in rotarod performance shown in figure 2?

We did not do that experiment, since the point was already made with the shorter CAG repeat expansion Genea 20 cells (48Q).

6. The rotarod data (Figure 5A) suggests a slight improvement with R6/2-hGPC treatment. However, the improvement is very subtle. Again, the extend and survival of engraftment should be carefully quantitatively characterized.

We have done so; the data are provided in **Table 1B**, and noted in the Results (**page 6**):

Striatal engraftment of the R6/2 mice by CD44-sorted hGPCs was robust (Figures 4A-B), and achieved densities of >15,000 human cells/mm³ by 16 weeks of age (Figure 4C and Table 1B), with substantial replacement of resident mouse HD astroglia with normal HTT-expressing human counterparts, as we have previously reported in wild-type murine hosts.

7. Mechanistically, the change of input resistance in MSNs is interesting. The authors showed that change of intrastriatal K⁺ levels. However, change of K⁺ concentration would only slightly change the driving force. This alone would not explain the dramatic change of input resistance that is observed here.

The change in neuronal input resistance is a dual function of MSN-intrinsic defects in potassium channel expression, in tandem with the deficits in astrocytic potassium buffering caused by diminished astrocytic potassium channel expression, as has been reported by Khakh and others.

Minor points:

Fig 3B: unclear that the hGPC-73Q group requires significantly fewer current injections to fire action potentials. To make it clearer at what current injection the MSNs first fire, the authors can slightly fade all other traces except the first 1-2 traces at which the MSNs fire.

We thank the referee for this suggestion. We have faded some of the traces, and the plot does indeed look much better, clearly showing that MSNs recorded in the 73Q hGPC-engrafted mice require fewer current injections to fire than do their 23Q hGPC-engrafted controls.

Figure 3G: an asterisk suggests that sEPSC frequency does decrease in the hGPC-73Q group. The text in the results (page 6) reads that sEPSC frequency does not change.

We thank the referee for pointing out this error. The difference was indeed not significant, as stated in the text. The asterisk was removed.

Some figure letters don't match the text (ie Fig 3), or are not formatted correctly (ie sup Fig 4).

Fixed.

On behalf of our group, I would like to again thank the referees for their time and effort, and for the many helpful comments that they offered on behalf of this manuscript, which has benefited greatly from their input.

My best,

Steven A. Goldman, MD, PhD

Reviewers' comments:

Reviewer #1 (Remarks to the Author):

Major claims of the paper:

To define the functional role of glia in Huntington disease, the authors apply a unique experimental strategy based on the development of a human HD glial chimeric mouse. The effect of huntingtin gene mutation in glia on motor performance and neuronal activity was assessed in mice by grafting human glial cells derived from both a hESC carrying huntingtin mutant and mHTT-transduced fetal forebrain glial precursor cells. To further support the role of glia in the development and progression of the HD pathology, the authors engrafted healthy human glia into R2/6 HD mice, partially rescuing motor performance and neuronal function, and extending the survival of the HD mouse model.

Novelty of the paper:

The paper is definitely novel. These well-designed experiments for addressing the role of glia in HD or other disease models are unique, advancing our understanding of the degenerative nature of HD.

Overall quality of the data:

The data is of high quality, with appropriate study designs. There are only a few minor comments to improve the final version of the manuscript.

Minor comments:

1. Figure 1B shows more expression of the hNT in the cortex, than in the striatum of the mouse.
2. Figure 1 is missing the annotation of the 'D' panel
3. The graph layout in Figure 2 suggests that all time points are significantly different between the animal groups, while Table S1 shows that this is not the case.
4. There is a mistake in referencing Figure 3D in the text of the results section, where it should be Figure 3E.
5. Please add statistical analysis or explain why in Figure 5 graph A, depicting multiple time points of analysis, there is no statistical analysis of time points 16 and 20 implicated in Table S3.
6. In the results section referring to the disease-associated hyperactivity of the R2/6 mice, it is suggested that the hGPC-graft attenuates all three phenotypes of the measured behavior: sniffing/scanning, locomotion and rearing. Please clarify that the graft affected two of these readouts.
7. In the results section referring to the age-dependent gait deficits of the R2/6 mice, stride duration is described, but not presented in Figure 5S. Additionally, the described significant effect on the body movement variability is not represented in the graph.
8. In the results section referring to the T-maze performance of the R2/6 mice, the graphs in figure 7S show time points 9 and 13 weeks, whereas the text mentions week 11. Please also be more clear in the text that both R2/6 mouse groups are significantly worse in their performance in all readouts at both time points when compared to the WT mice, but the R2/6 hGPC grafted mice were significantly better compared to the R2/6 mice.
9. Please fix the graph legends in figure S4.
10. In the discussion in the 1st paragraph, delete the word 'glia' after 'glia (23Q)'.

Influence of the paper on the field:

This paper has a potential to be highly influential in the field.

Statistics:

Statistics are of high quality. The only comment regards the statistical analysis done on the behavioral readouts in the R2/6 mice: the description of the statistical analysis is not clear regarding which analyses were applied to the data in figure 6 and supplementary figures 5 and 6.

Reviewer #2 (Remarks to the Author):

The work of Benraiss et al. addresses significant questions in stem cell biology and transplants. The work is original and of general interest. They evaluate whether transplantation of glia offers therapeutic benefit in mouse models of HD. The authors demonstrate that glia derived from human

ESC cells improve rotarod, had decreased striatal atrophy and improved measures with the SmartCube. The effects on lifespan are modest. They also engrafted glia from human expressing mutant HTT (derived from ESCs) and demonstrated significant contribution to disease pathology and behavioral changes. The strength of the investigation is the significant questions addressed. There are some points that should be further clarified.

(1) It is not clear if the same production and purification of CD-44-defined astroglial progenitors and bipotential oligodendrocyte-astrocyte was used for the experiments shown in the work. The manuscript reads as if they were distinct. If each preparation was done differently than it may not be that glia positive for expansion promotes disease and those negative are neuroprotective. It could be how the cells were made. This needs further clarification. Use of GENE20 and GENE 19 would make the studies consistent.

(2) Why is the electrophysiology done in Figure 3A-B with a lentivirus mHTT transduced system? Was it not sufficient to transplant the CD-44 positive astrocytes from the GENE with 48 repeats and obtain alterations in electrophysiology? Why not report this if that is the cause. The shift in systems and rationale seem out of place.

(3) Could the authors in the material methods present a table of mice transplanted per study and how they were used? It seems like a separate set where made to ship to Psychogenics, Inc. Did this set have survival measured as well? Was there any effect on weight in the studies? Did the response occur in both females and males?

(4) The transplantation is done 24 hours after birth and this does not seem like it directly tests the whether these cells can be used therapeutically in HD. Do the authors have benefit if the transplantation is done when the mice are adults?

(5) Did two separate studies of the transplanted cells into mice give similar trends?

(6) Is Table 1 described in the results section?

(7) Figure 2 does give a sense of the variability in rotarod for the different mice as plotted.

(8) Some figures have the number of animals used while others do not. Perhaps specify in each figure.

Reviewer #3 (Remarks to the Author):

Manuscript #: NCOMMS-16-01163-T

Title: Human glia can both induce and rescue aspects of disease phenotype in Huntington Disease

Comments:

In this manuscript, the authors try to examine the role of glia in Huntington's disease (HD). In a previous study, the authors established a protocol to generate Glial Progenitor cells (GPCs) and their derived astroglia and oligodendrocytes from human embryonic stem cells (hESCs). In this work, the authors try to explore whether neonatal engraftment of these cells into immunodeficient mice could have therapeutic effect in mouse models of HD. Stem cell therapy has big potential for treating neurodegenerative diseases. Previous studies have attempted to use derived neuron engraftment for treating HD. In this study, the authors use a chimeric mouse where control and mutant GPCs generated from human embryonic stem cells are engrafted in the striata of immunodeficient or R6/2 mice. Using a combined behavioral and electrophysiological approach, the authors suggest that glial pathology contributes to HD by increasing interstitial potassium in the striatum. The mechanism still remains unknown. But the authors showed that glial regulation of extracellular potassium level is involved. Overall, the observation is interesting. The current study seems to be preliminary. A few important issues need to be addressed.

Major issues:

1. "data not shown" appears several places. In general, this statement does not help. The authors should either show the data, or eliminate such statement throughout the manuscript.

2. The extend of GPCs engraftment is not well-characterized. The authors focus on the striatum, which is the primary region for neurodegeneration in HD. However, in their studies, the cortical regions are also largely affected. It is known that pathophysiological changes in cortex also contribute to HD.

Therefore, it is critical to perform controls. GPCs engraftment should be well controlled in striatum alone, cortex alone, etc. In addition, the authors should also take into consideration of the extend of engraftment, region, cell counts, etc. It is critical to perform experiments with high rigor in order to reach conclusion about causal effect.

3. Fig 1A: it would be more ideal to compare multiple consecutive sections taken at similar locations (the olfactory bulb is present in some sections but not others). The way it is currently presented implicates differences in spread between the experimental groups. Does the extend of engraftment correlate with the behavioral outcome?

4. Tumorigenesis is generally a big potential problem for stem cell therapy. The authors did indicate that no tumor was observed in all studies. However, this should be taken into more serious consideration for in vivo studies. Tumor markers should be examined using IHC.

5. Does fetal forebrain hESC studies (hGPC-23Q and hGPC-73Q) engraftment also cause similar decline in rotarod performance shown in figure 2?

6. The rotarod data (Figure 5A) suggests a slight improvement with R6/2-hGPC treatment. However, the improvement is very subtle. Again, the extend and survival of engraftment should be carefully quantitatively characterized.

7. Mechanistically, the change of input resistance in MSNs is interesting. The authors showed that change of intrastriatal K^+ levels. However, change of K^+ concentration would only slightly change the driving force. This alone would not explain the dramatic change of input resistance that is observed here.

Minor points:

Fig 3B: unclear that the hGPC-73Q group requires significantly fewer current injections to fire action potentials. To make it clearer at what current injection the MSNs first fire, the authors can slightly fade all other traces except the first 1-2 traces at which the MSNs fire.

Figure 3G: an asterisk suggests that sEPSC frequency does decrease in the hGPC-73Q group. The text in the results (page 6) reads that sEPSC frequency does not change.

Some figure letters don't match the text (ie Fig 3), or are not formatted correctly (ie sup Fig 4).